# Driving factors of conifer regeneration dynamics in eastern Canadian boreal old-growth forests

Maxence Martin[1,2]*, Miguel Montoro Girona[2,3,4], Hubert Morin[1,2]

**1** Département des Sciences Fondamentales, Université du Québec à Chicoutimi, Chicoutimi, Canada, **2** Centre d'étude de la forêt, Université du Québec à Montréal, Montréal, Canada, **3** Institut de Recherche sur les Forêts, Université du Québec en Abitibi-Témiscamingue, Amos, Canada, **4** Restoration Ecology Group, Department of Wildlife, Fish, and Environmental Studies, Swedish University of Agricultural Sciences (SLU), Umeå, Sweden

* maxence.martin1@uqac.ca

## Abstract

Old-growth forests play a major role in conserving biodiversity, protecting water resources, and sequestrating carbon, as well as serving as indispensable resources for indigenous societies. Novel silvicultural practices must be developed to emulate the natural dynamics and structural attributes of old-growth forests and preserve the ecosystem services provided by these boreal ecosystems. The success of these forest management strategies depends on developing an accurate understanding of natural regeneration dynamics. Our goal was therefore to identify the main patterns and drivers involved in the regeneration dynamics of old-growth forests with a focus on boreal stands dominated by black spruce (*Picea mariana* (L.) Mill.) and balsam fir (*Abies balsamea* (L.) Mill.) in eastern Canada. We sampled 71 stands in a 2 200 km$^2$ study area located within Quebec's boreal region. For each stand, we noted tree regeneration (seedlings and saplings), structural attributes (diameter distribution, deadwood volume, etc.), and abiotic (slope and soil) factors. The presence of seed-trees located nearby and slopes having moderate to high angles most influenced balsam fir regeneration. In contrast, the indirect indices of recent secondary disturbances (e.g., insect outbreaks or windthrows) and topographic constraints (slope and drainage) most influenced black spruce regeneration. We propose that black spruce regeneration dynamics can be separated into distinct phases: (i) layering within the understory, (ii) seedling growth when gaps open in the canopy, (iii) gradual canopy closure, and (iv) production of new layers once the canopy is closed. These dynamics are not observed in paludified stands or stands where balsam fir is more competitive than black spruce. Overall, this research helps explain the complexity of old-growth forest dynamics, where many ecological factors interact at multiple temporal and spatial scales. This study also improves our understanding of ecological processes within primary old-growth forests and identifies the key factors to consider when ensuring the sustainable management of old-growth boreal stands.

**Data Availability Statement:** Data are available from Figshare (DOI: 10.6084/m9.figshare.12557708.v1).

**Funding:** HM: "Chaire de recherche industrielle du CRSNG sur la croissance de l'épinette noire et

l'influence de la tordeuse des bourgeons de l'épinette sur la variabilité des paysages en zone boréale,", founded by the Natural Sciences and Engineering Research Council (NSERC) of Canada, the Canada Research Industrial Chairs Program (no grant number; https://www.nserc-crsng.gc.ca/) MMG: professor start-up fund at University of Québec in Abitibi Temiscamingue (UQAT; no grant number; https://www.uqat.ca/) and silvicultural research grant (MRC-Abitibi; no grant number; https://mrcabitibi.qc.ca/).

**Competing interests:** The authors have declared that no competing interests exist.

## Introduction

The global extent of primaryaold-growth forest has declined markedly over the past few centuries through the cumulative and increasing impact from anthropogenic activities within these forest landscapes [1–3]. The boreal forest, most of which is situated in Canada and Russia, is currently the largest reserve of natural forest on our planet [3]. Boreal old-growth forest has also experienced rapid loss over the last centuries [1,4,5]. The remaining old-growth forests are critically important to biodiversity, water resources, carbon sequestration and storage, and these stands remain integral elements of indigenous societies and even human health [3,6]. The sustainable management of boreal forests has a primary goal of protecting the remaining old-growth forests. Restoring the integrity of intact forests is also an urgent issue; this is especially true in Fennoscandia where old-growth forests have been almost completely eliminated [7]. We are therefore facing a critical situation where novel silvicultural practices and restoration strategies are now priorities in the context of the global biodiversity crisis, climate change, and forest sustainability.

Effective forest restoration strategies require an accurate understanding of the natural dynamics of old-growth forests. Tree regeneration is an essential process in forest ecosystems to ensure the persistence and resilience of forest stands when subjected to various disturbances [8,9]. Forest science is therefore placing increased importance on understanding tree regeneration following natural and anthropogenic disturbances (e.g., [10–16]). Nonetheless, regeneration dynamics in old-growth forests remain an understudied subject in ecology; this absence is particularly true for the boreal biome. Moreover, because of the scarcity of old-growth stands in many boreal regions, conducting studies related to this subject is often challenging, given the lack of reference sites. This need for baseline data underscores the important scientific value of the boreal biome in eastern Canada where some regions still contain large intact stands of forest as intensive forest management practices began only relatively recently, i.e., since the 1960s [17,18]. The study of regeneration dynamics in the boreal old-growth forests of eastern Canada can thus benefit all boreal regions, especially areas where these ecosystems have been almost completely eliminated.

Black spruce (*Picea mariana* (L.) Mill.) and balsam fir (*Abies balsamea* (L.) Mill.) are the two main late-successional species in the eastern Canadian boreal forest [19]. Pure black spruce or mixed black spruce–balsam fir stands are the most common old-growth forest types in eastern Canada [19–21]. Old-growth forests are also, however, the most logged forest type in this territory, leading to the rapid loss of old-growth forest surfaces [5,22,23]. Pure black spruce stands are under even greater pressure as this specific old-growth forest type is most selected for logging given the high economic value of this species [23].

Both black spruce and balsam fir are well adapted to long (>150 years) periods of suppressed growth in the understory [24–26]. These species are also able to regenerate under their own cover, mostly through vegetative reproduction for black spruce—regeneration by layers—and sexual reproduction, i.e., seed origin, for balsam fir [19]. Previous studies have highlighted that the seedling densities of black spruce and balsam fir are similar under gaps or canopy cover [27–29]. Where a gap in the canopy opens as a result of a secondary disturbance (e.g., insect outbreak or windthrow), the gap-fillers are generally pre-established regeneration rather than seeds or layers, which become established following the disturbance [30,31]. Once a gap is created, the regeneration trees of both species increase their vertical growth to reach the overstory relatively quickly [26,32,33]. However, black spruce and balsam fir differ in their ecological strategies in terms of growth, sensitivity to primary and secondary disturbance, resistance to fire, and seed dispersal; thus, these differences should vary their respective regeneration dynamics. Balsam fir regeneration is seen as being more competitive than that of black

spruce owing to balsam fir seedlings' faster and more intense growth response to canopy openings [31,34]. Balsam fir, however, is more vulnerable to spruce budworm (*Choristoneura fumiferana* (Mills.)) outbreaks, windthrow, and root rot than black spruce [35–38]. Moreover, balsam fir seeds are not adapted to fire, making this species strongly dependent on the proximity of seed-trees, as opposed to black spruce, which is very well adapted to fire events [39]. Black spruce also outcompetes balsam fir on wet soils [39].

From the abovementioned observations, stands in the old-growth forests of eastern Canada are expected to shift between black spruce–dominated stands and black spruce–balsam fir mixed stands over time [21,28,40]. As well, the structure of these stands varies over time (decades and centuries), even though tree species' composition remains the same [40,41]. At a decennial scale, it is therefore likely that the characteristics of the understory, e.g., tree density or tree species composition within the regeneration layer, will change significantly and rapidly because of the succession of tree-mortality and canopy-closure phases.

Understanding the regeneration process in old-growth forests is therefore critical for developing management strategies and silvicultural treatments that minimize differences between managed and unmanaged forests [42]. Our study objective is to identify the main patterns and factors involved in the regeneration dynamics of black spruce and balsam fir in eastern Canadian boreal old-growth forests under natural secondary disturbance regimes. Therefore, we do not consider natural stand-replacing disturbances (crown fires) or anthropogenic disturbances (logging) in this study. We hypothesize that (1) for both black spruce and balsam fir, sapling density will increase in relation to the secondary disturbance–related structural changes, such as an opening of the canopy or an increase in deadwood volume, and (2) the main differences between the general patterns of regeneration dynamics for black spruce and balsam fir are due to abiotic constraints and the availability of proximal balsam fir seed-trees.

## Materials and methods

### Ethics statement

This study was carried out in Quebec's public forests and outside any protected area. Quebec's Ministry of Forests, Wildlife and Parks, the governmental authority responsible for these areas, gave us permission to take samples from trees when necessary. No permit was required to conduct the field surveys. This research did not involve endangered or protected species

### Study area

Our study involved a 2 200 km$^2$ region of public forest southeast of Lake Mistassini, Quebec, Canada (Fig 1) within an area extending between 50˚07´23″N to 50˚30´00″N and 72˚15´00″W to 72˚30´00″W. The study zone is crossed by the Mistassini, Ouasiemsca, and Nestaocano rivers and lies within the western subdomain of the black spruce–feather moss bioclimatic domain [43]. Regional climate is subarctic with a short growing season (120–155 days). Mean annual temperature ranges between −2.5 and 0.0˚C, and mean annual precipitation varies from 700 to 1000 mm [43]. Surficial deposits consist mainly of thick glacial till, forming a low-lying topography characterized by gentle hills that vary in altitude from 350 to 750 m asl [44]. Black spruce and balsam fir dominate the stands across this territory, and jack pine (*Pinus banksiana* Lamb.), white spruce (*Picea glauca* (Moench) Voss), paper birch (*Betula papyrifera* Marsh.), and trembling aspen (*Populus tremuloides* Michx.) are the secondary tree species.

Fire is the main driver of natural stand-replacing disturbances on this territory [45], whereas spruce budworm outbreaks are the principal agent of secondary disturbance [26]. This territory was unmanaged until 1991 when intensive timber exploitation began, mostly by

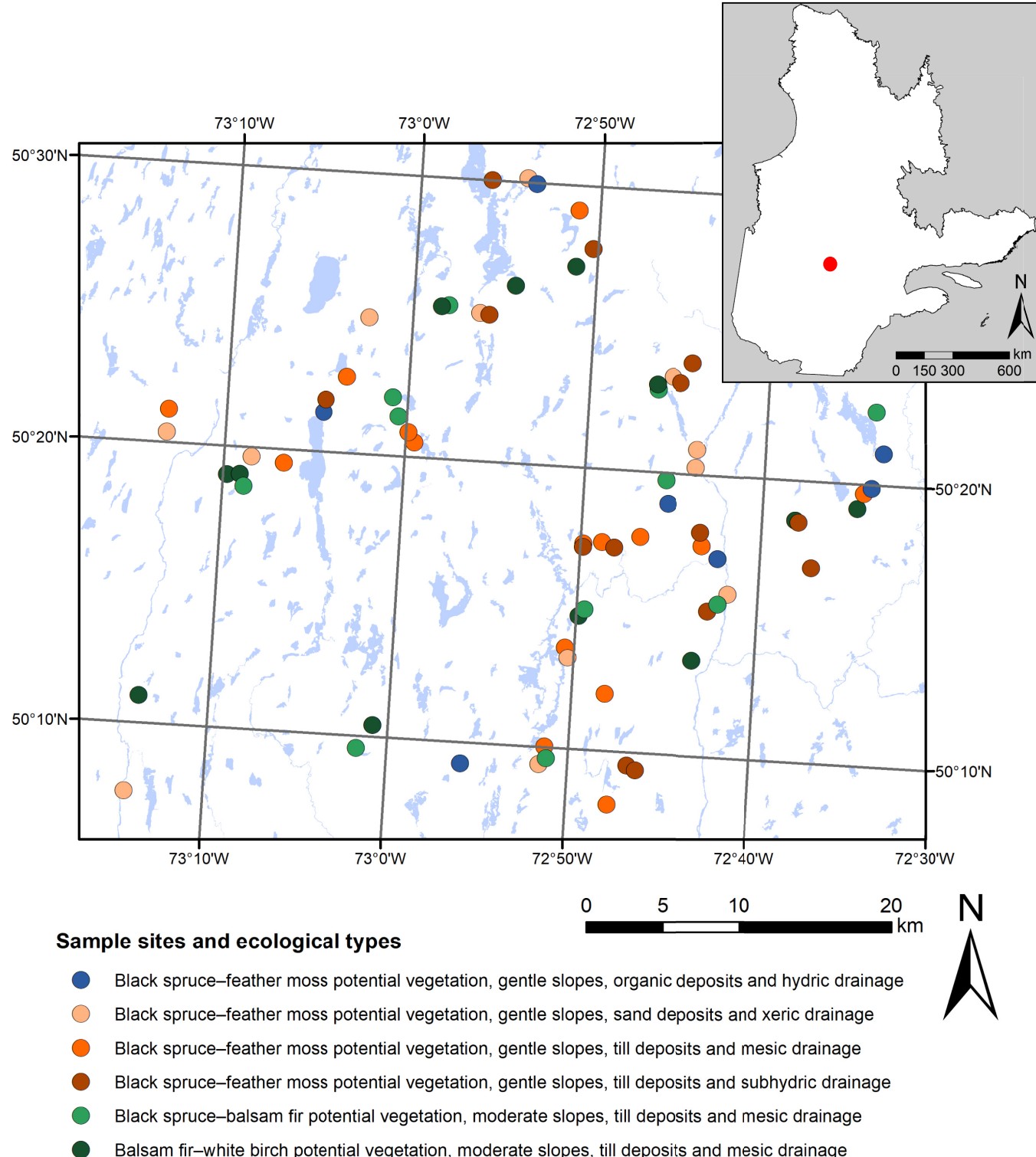

**Sample sites and ecological types**

- 🔵 Black spruce–feather moss potential vegetation, gentle slopes, organic deposits and hydric drainage
- 🟠 Black spruce–feather moss potential vegetation, gentle slopes, sand deposits and xeric drainage
- 🟠 Black spruce–feather moss potential vegetation, gentle slopes, till deposits and mesic drainage
- 🟤 Black spruce–feather moss potential vegetation, gentle slopes, till deposits and subhydric drainage
- 🟢 Black spruce–balsam fir potential vegetation, moderate slopes, till deposits and mesic drainage
- 🟢 Balsam fir–white birch potential vegetation, moderate slopes, till deposits and mesic drainage
- 🟦 Water bodies

**Fig 1. Map of the study territory showing the location of the sample sites (red filled circles).** The insert map indicates the location of the study territory in the province of Quebec, Canada (red dot). The cartographic data used to generate this comes from freely available maps map ("*Carte écoforestière du troisième inventaire*" and "*Classification écologique du territoire québécois*"), published by the Government of Quebec (https://www.donneesquebec.ca/fr/). This map was produced with the ArcGIS Desktop software, version 10.7.1.

clearcutting. The harvested area remained relatively low until 2000; harvesting increased significantly after this date.

## Experimental design

We sampled 71 stands in the study area, in either 2015 or 2016, and applied a stratified random sampling approach. All surveyed stands were primary forests and undisturbed by human activities, such as logging. Site selection considered two main criteria: (1) that sites reflected the six dominant "ecological types" found within the study area, according to the ecological classification of the Quebec Ministry of Forests, Wildlife and Parks (MFWP) [43], and (2) that sites must contain two minimal stand-age classes (80–200 years and >200 years). Ecological types are defined through a combination of site potential vegetation, slope classes, surface deposits, and drainage classes. The six dominant MFWP ecological types covered more than 72% of the forested area on the study territory. They included: (1) balsam fir–white birch potential vegetation having moderate slopes, till deposits, and mesic drainage; (2) black spruce–balsam fir potential vegetation having moderate slopes, till deposits, and mesic drainage; (3) black spruce–feather moss potential vegetation (BSFM) having gentle slopes, sand deposits, and xeric drainage; (4) BSFM having gentle slopes, till deposits, and mesic drainage; (5) BSFM having gentle slopes, till deposits, and subhydric drainage; and (6) BSFM having gentle slopes, organic deposits, and hydric drainage.

The age classes correspond to the successional stages of the transition process toward the old-growth stage in Quebec boreal forests [20,46,47]: 80–200 years (beginning of the transition toward an old-growth forest) and >200 years (end of the transition to an old-growth forest). However, data from aerial forest inventories conducted by the Quebec government did not provide sufficient resolution to discriminate stands older or younger than 200 years. We therefore used GIS software to identify old stands (estimated age >80 years) in the study area corresponding to the target potential vegetation. In addition, we chose only those polygons for which one of the stand edges fell within 200 m of a road. We conducted preliminary surveys of the stands, during which we collected cores from the root collar of five dominant or codominant trees per site. We determined from tree-ring counts of these cores using a binocular microscope.

As the study area is very remote and has limited road access, we added additional logistical criteria to the site selection process; for example, we sampled only sites that were accessible via the existing road network. As well, our surveys were systematically placed at 125 m from the stand edge to limit the influence of the edge effect. As a result of the various constraints detailed above, the final selection of sample stands reflected the availability and accessibility of the different stand types in the study area.

## Plot measurements

At each site, we established a permanent square plot (400 m$^2$) as the basis for all subsequent transects and subplots (Fig 2). For each plot, we sampled all merchantable trees—trees having a diameter at breast height (DBH) ≥9 cm—in each 400-m$^2$ plot. The sampled attributes were species, DBH, and vitality (alive or dead). We then surveyed all saplings—stems having a DBH <9 cm and height ≥1.3 m—in two 100-m$^2$ (10 m × 10 m) subplots within the larger plot (Fig

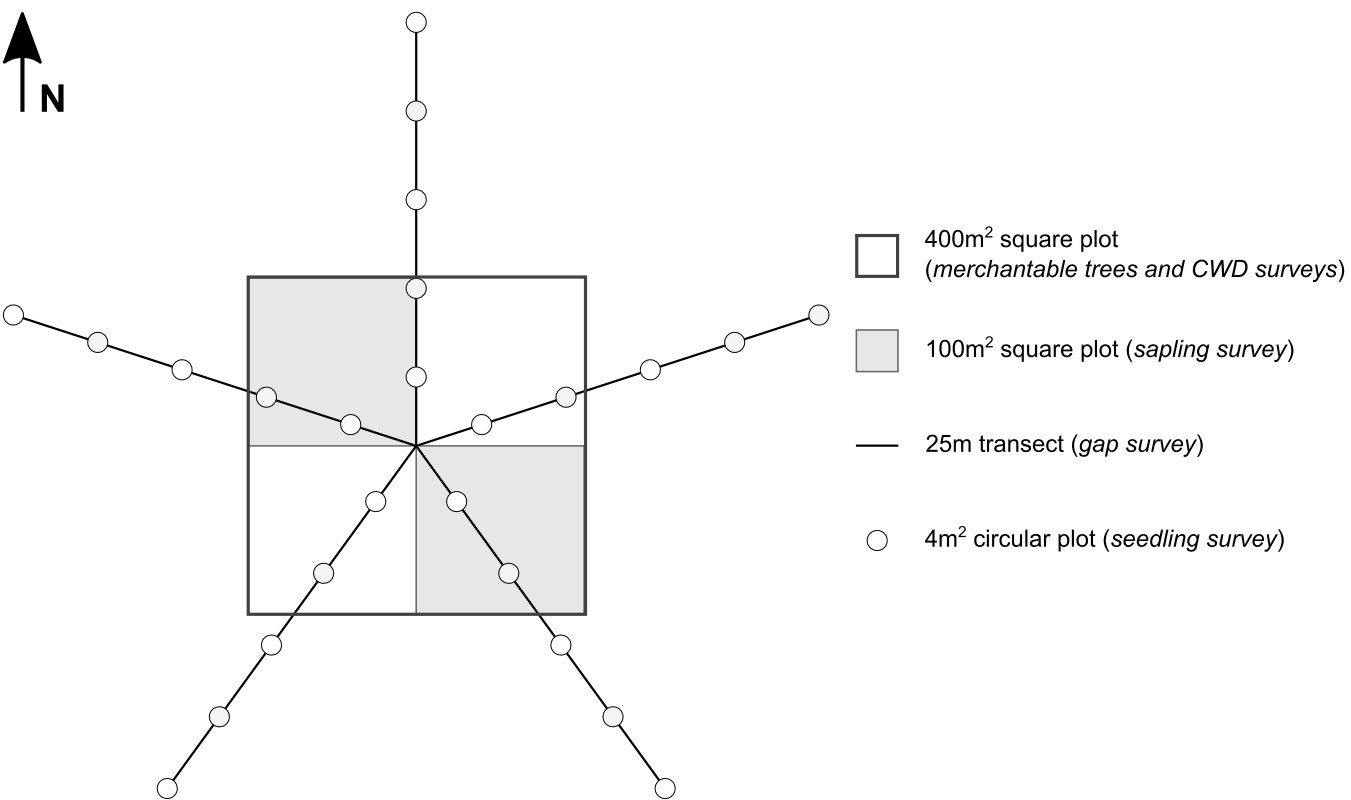

**Fig 2. Schematic representation of the experimental design used for the sample sites.** N: north; CWD: coarse woody debris.

2). The sampled attributes for saplings were species and DBH. To count seedlings and quantify their attributes, we established twenty-five 4-m$^2$ circular plots along five 25-m-long transects (5 circular plots/transect) that extended out from the center of the 400-m$^2$ plot. The angle between two neighboring 25 m-long transects was equal to 72°. Transect 1 was the transect oriented due north. Along a transect, the first circular plot was placed 5 m from the center of the 400-m$^2$ plot, with the following circular plots each separated by 5 m. In each 4-m$^2$ plot, we inventoried all seedlings by tree species. We also measured gap length along the five 25-m-long seedling transects. We defined gaps as all sections along the transect where the canopy was less than two-thirds height of the dominant trees [28] and representing a space greater than 2 m. We included this second criterion to avoid confusion between actual gaps and the natural separation between tree crowns in these forests [40]. We defined the size of our study from other similar studies and the forest survey methods of the Quebec provincial government [15,48].

In addition to these seedling transects, we surveyed coarse woody debris along four 20-m-long transects that followed the edges of the 400-m2 plot. We surveyed the diameter of any coarse woody debris intersecting with the transect. We recorded this information for only debris having a diameter >9 cm at the transect intersection. Debris items buried in the organic layer at a depth >15 cm were not sampled. We determined the site's soil type by digging a soil profile at the center of the 400-m2 plot. To determine the topography of the plot, we used a clinometer to measure slope. All sampled data were collected in the same year for each plot.

## Data compilation

We applied the following equation to estimate regeneration attributes, i.e., seedling and sapling density, for black spruce and balsam fir:

$$D = \sum_{i=1}^{n} R \times \frac{10\,000}{\sum_{i=1}^{n} S}$$

where $D$ corresponds to the density per hectare, $R$ is the number of seedlings or saplings sampled in each of the $n$ plots surveyed, and $S$ represents the surface (in $m^2$) of each of the $n$ plots.

Martin et al. [40] had previously computed several structural and environmental attributes for each of the sampled sites used in this study (Table 1). Some of these attributes relate to stand structure, including merchantable tree density, basal area, Weibull's shape parameter of diameter distribution [49], and gap fraction, i.e., the ratio between gap length and total transect length, sensu Battles et al. [50]. Other attributes relate to stand composition, such as the basal area proportion of balsam fir. For estimating deadwood, Martin et al. [40] computed the volume of coarse woody debris per hectare using the formula of Marshall et al. [51]; however for this study, we also calculated the basal area of snags, i.e., merchantable dead trees at each study site, an attribute absent from the earlier Martin et al. [40] study. We evaluated forest succession

**Table 1. Description of the regeneration, stand structure, and abiotic attributes measured at the study sites, adapted from Martin et al. [40].**

| Category | Attribute | Unit | Description |
|---|---|---|---|
| Regeneration | Black spruce seedling density | n/ha | Number of living black spruce seedlings per hectare |
| | Black spruce sapling density | n/ha | Number of living black spruce saplings per hectare |
| | Balsam fir seedling density | n/ha | Number of living balsam fir seedlings per hectare |
| | Balsam fir sapling density | n/ha | Number of living balsam fir saplings per hectare |
| Stand structure | Tree density* | n/ha | Number of living merchantable stems per hectare |
| | Basal area* | $m^2$/ha | Basal area of the living merchantable trees per hectare |
| | Balsam fir proportion* | % | Proportion of balsam fir in the basal area |
| | Coarse woody debris volume* | $m^3$/ha | Coarse woody debris volume per hectare |
| | Snag basal area | $m^2$/ha | Basal area of the dead merchantable trees per hectare |
| | Gap fraction* | % | Mean percentage of the canopy under gaps |
| | Stand height* | m | Mean height value of the dominant trees sampled at each site |
| | Weibull's shape parameter* | - | Weibull's function shape parameter (WSP, Bailey and Dell 1973) based on the diameter distribution of saplings and merchantable trees. A WSP of ≥1.5 represents a Gaussian distribution of the diameters, 1 ≤ WSP < 1.5 reflects an irregular distribution, and WSP <1 describes a reverse J-shaped distribution |
| | Cohort basal area proportion* | - | Replacement index of the even-aged cohort by old-growth cohorts, as defined by Kneeshaw and Gauthier (2003), and values range from 0 to 1. CBAP = 0 indicates a stand having a single even-aged cohort, and CBAP = 1 indicates a stand where old-growth cohorts have replaced all of the even-aged cohort |
| Abiotic | Minimum time since last fire* | years | Age of the oldest tree |
| | Slope | % | Mean slope value within the 400-$m^2$ plot |
| | Organic horizon thickness* | cm | Mean thickness of the organic horizon of the soil profile |

"*" indicates attributes computed by Martin et al. [40].

from the minimum time since the last fire, i.e., the age of the oldest tree sampled, and the cohort basal area proportion (CBAP; sensu [52]). The latter attribute is an indicator of the stand transition from an even-aged to old-growth stage, i.e., the stage where almost all trees of the first cohort following the last stand-replacing disturbance have disappeared. A CBAP $\approx 0$ indicates a stand where all trees belong to the first cohort, and a CBAP = 1 indicates a stand where the first cohort has been replaced entirely by a new shade-tolerant cohort. Finally, we detailed the topographic and pedologic characteristics of the studied stands using slope and the depth of the organic horizon, respectively.

## Data analysis

First, we performed k-means clustering [53] on black spruce and balsam fir regeneration attributes to identify the main patterns driving the regeneration dynamics of these two tree species in eastern Canadian boreal old-growth forests. To highlight the differences between the two species, we ran k-means clustering separately for each species. The clustering of black spruce regeneration relied on black spruce seedling and sapling densities of all 71 sites. Similarly, clustering of balsam fir regeneration also relied on balsam seedling and sapling densities; however, balsam fir seedlings and saplings were absent in 24 sites. We thus removed these sites for the clustering of balsam fir (47 plots remaining) to eliminate any influence from sites lacking this balsam fir regeneration. For each cluster analysis, we determined the optimal number of regeneration clusters using the simple structure index (SSI; [54]) criterion. The highest SSI value indicates the optimal k-means clustering partition. Separately for both species, we compared regeneration as well as the structural and environmental attributes within the clusters. We used analysis of variance (ANOVA) when the ANOVA conditions were fulfilled (data normality and homoscedasticity) or Kruskal-Wallis nonparametric analysis of variance when these conditions were not met. When ANOVA or the Kruskal-Wallis tests were significant, we performed a Tukey post hoc test [55] or a Fisher's least significant difference test [56], respectively. Moreover, we also calculated Spearman's rank correlation coefficient between the regeneration and structural/environmental attributes. This analysis served to provide valuable information for interpreting our results by highlighting the strength of the relationship between regeneration and these various attributes. As a complement to Spearman's rank correlation, we also performed bootstrapped linear regression to calculate the relative importance of the structural and environmental attributes for black spruce and balsam fir regeneration, following the methodology of Lindeman et al. [57]. For each regeneration attribute (i.e., black spruce or balsam fir, seedling or sapling), we used only the structural and environmental attributes found to be significant in the Spearman's rank correlation test, for which we ran 10 000 bootstrapping iterations.

Given that the stands were not monitored over time, we accounted for the combinations of particular structural attributes as indicators of the age and severity of disturbance. For example, we consider a high gap fraction, a large volume of woody debris, and a small basal area to indicate a recent disturbance of low to moderate severity [58]. Nonetheless, the exact cause of the disturbance (e.g., insect outbreak or windthrow) remains impossible to determine. Similarly, we considered the slope and thickness of organic matter in the soil as respective indicators of site topography and drainage. All analyses were performed using R software, version 3.3.1 [59] and the *vegan* [60], *Hmisc* [61], *relaimpo* [62] and *agricolae* [63] packages, applying a *p*-threshold of 0.05.

## Results

### Black spruce and balsam fir regeneration

For the cluster analysis of black spruce regeneration, we determined eight clusters to be optimal (SSI = 2.23; Fig 3). Black spruce seedling and sapling densities differed significantly

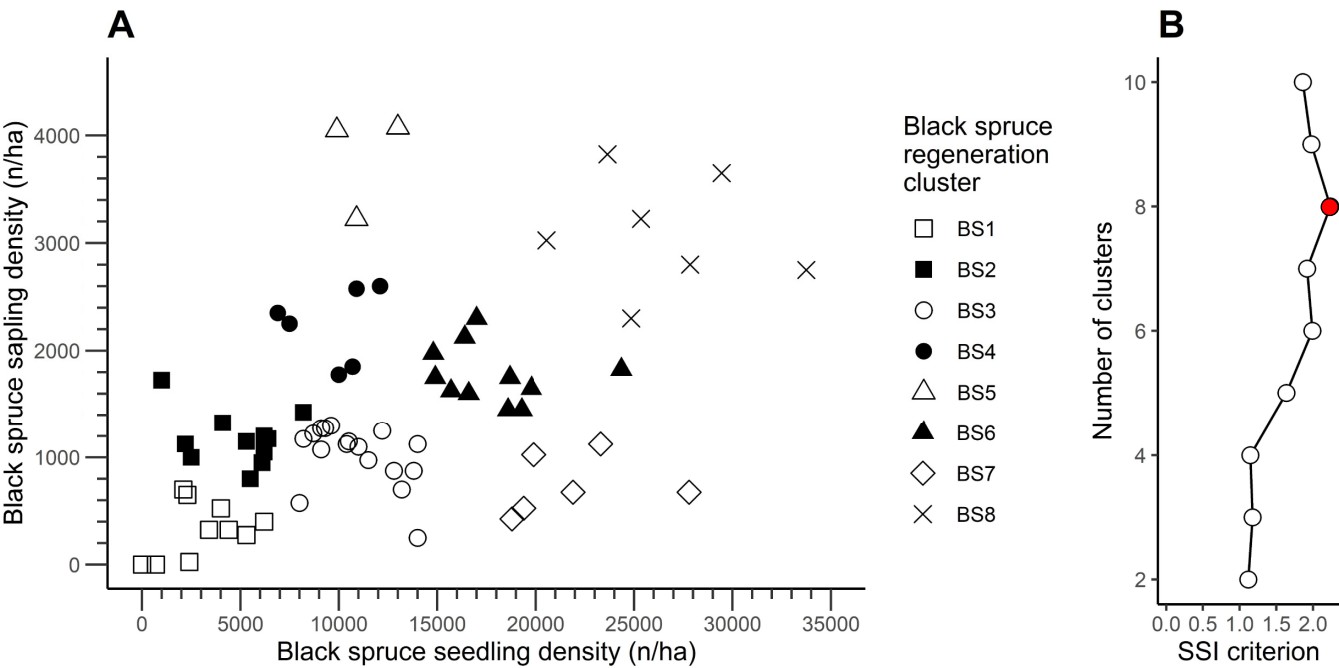

**Fig 3.** (A) Density of black spruce seedlings and saplings at the 71 studied sites, grouped by black spruce regeneration clusters. (B) Value of the SSI criterion according to the number of clusters for black spruce, using k-means clustering. Filled circle in (B) indicates the highest value of the SSI criterion.

between the black spruce regeneration clusters (BS; Table 2A). Black spruce seedling density was more than 8× higher in cluster BS8, having the highest density (26 543 seedlings/ha), than in cluster BS1, characterized by the lowest seedling density values (3 008 seedlings/ha). Black spruce seedling density did not differ between clusters BS4, BS5, and BS6. Regarding the density of black spruce saplings, cluster BS1—having the lowest values at 322 saplings/ha—contained a sapling density 12× lower than that of cluster BS5, which had the highest density of black spruce seedlings at 3 783 saplings/ha. The remaining clusters, characterized by intermediate values of black spruce sapling density, aligned along a gradient. We also observed significant differences in balsam fir seedling density between clusters. For balsam fir seedling density within the clusters of black spruce regeneration, we observed significant differences, ranging from 873 seedlings/ha (lowest value, cluster BS7) to 9 720 seedlings/ha (highest value, cluster BS1); however, balsam fir sapling density did not differ significantly between the clusters.

For balsam fir regeneration, two and four clusters produced an identical SSI criterion value of 1.14 (Fig 4). Nonetheless, to obtain a more detailed evaluation of the dynamics of balsam fir regeneration, we chose to use four clusters (BF; Table 2B). Balsam fir seedling and sapling density varied markedly between clusters, and we identified significant differences between the clusters for every attribute. For example, the density of balsam fir seedlings within cluster BF4, marked by the highest seedling density at 18 740 seedlings/ha, was almost 20× that of the cluster having the lowest density of balsam fir seedlings (957 seedlings/ha; cluster BF1). Similarly, the highest density of balsam fir saplings (7 442 saplings/ha; cluster BF3) was 33× that of the cluster having the lowest density (223 saplings/ha; cluster BF1). Differences between clusters in terms of black spruce seedling or sapling density were less striking, although both attributes differed significantly between the clusters. Black spruce seedling density varied from 1 900 to 14 379 seedlings/ha, whereas black spruce sapling density ranged from 470 to 1 670 saplings/ha (clusters BF4 and BF1, respectively, for both cases).

**Table 2. Mean and standard deviation of the regeneration attributes for (A) black spruce regeneration clusters and (B) balsam fir regeneration clusters.**

**A: Black spruce regeneration**

| Cluster | BS1 ($n$ = 10) | BS2 ($n$ = 11) | BS3 ($n$ = 17) | BS4 ($n$ = 6) | BS5 ($n$ = 3) | BS6 ($n$ = 11) | BS7 ($n$ = 6) | BS8 ($n$ = 7) |
|---|---|---|---|---|---|---|---|---|
| Black spruce seedling density (n/ha) | 3 080 ± 1 959 e | 4 882 ± 2 177 d | 10 906 ± 2 096 c | 9 683 ± 2 048 c | 11 267 ± 1 582 c | 17 836 ± 2 777 b | 21 850 ± 3 365 a | 26 543 ± 4 295 a |
| Black spruce sapling density (n/ha) | 322 ± 257 f | 1 175 ± 251 d | 1 019 ± 286 d | 2 233 ± 353 bc | 3 783 ± 484 a | 1 773 ± 269 c | 742 ± 277 e | 3 082 ± 532 ab |
| Balsam fir seedling density (n/ha) | 9 720 ± 7 920 a | 5 773 ± 7 621 ab | 1 835 ± 4 253 c | 1 917 ± 4 224 c | 3 333 ± 5 687 ac | 873 ± 2 039 c | 6 317 ± 8 655 ab | 2 357 ± 5 474 bc |
| Balsam fir sapling density (n/ha) | 1 492 ± 1 499 | 2 516 ± 3 374 | 228 ± 495 | 500 ± 765 | 608 ± 1 032 | 125 ± 357 | 592 ± 668 | 200 ± 416 |

**B: Balsam fir regeneration**

| Cluster | BF1 ($n$ = 28) | BF2 ($n$ = 11) | BF3 ($n$ = 3) | BF4 ($n$ = 5) | | | | |
|---|---|---|---|---|---|---|---|---|
| Black spruce seedling density (n/ha) | 14 379 ± 7 824 a | 10 773 ± 9 688 ab | 1 900 ± 794 b | 9 180 ± 7 258 ab | | | | |
| Black spruce sapling density (n/ha) | 1 670 ± 1 025 a | 1 282 ± 1 131 ab | 1 283 ± 388 ab | 470 ± 151 b | | | | |
| Balsam fir seedling density (n/ha) | 957 ± 1 520 c | 9 827 ± 3 243 b | 1 6267 ± 5 460 ab | 1 8740 ± 2 756 a | | | | |
| Balsam fir sapling density (n/ha) | 223 ± 397 c | 1 464 ± 683 b | 7 442 ± 1 934 a | 2 590 ± 1 066 ab | | | | |

Different letters indicate significant differences at $p < 0.05$, following a > b > c > d > e. BS: black spruce; BF: balsam fir. All the analyses were performed using Kruskal-Wallis tests.

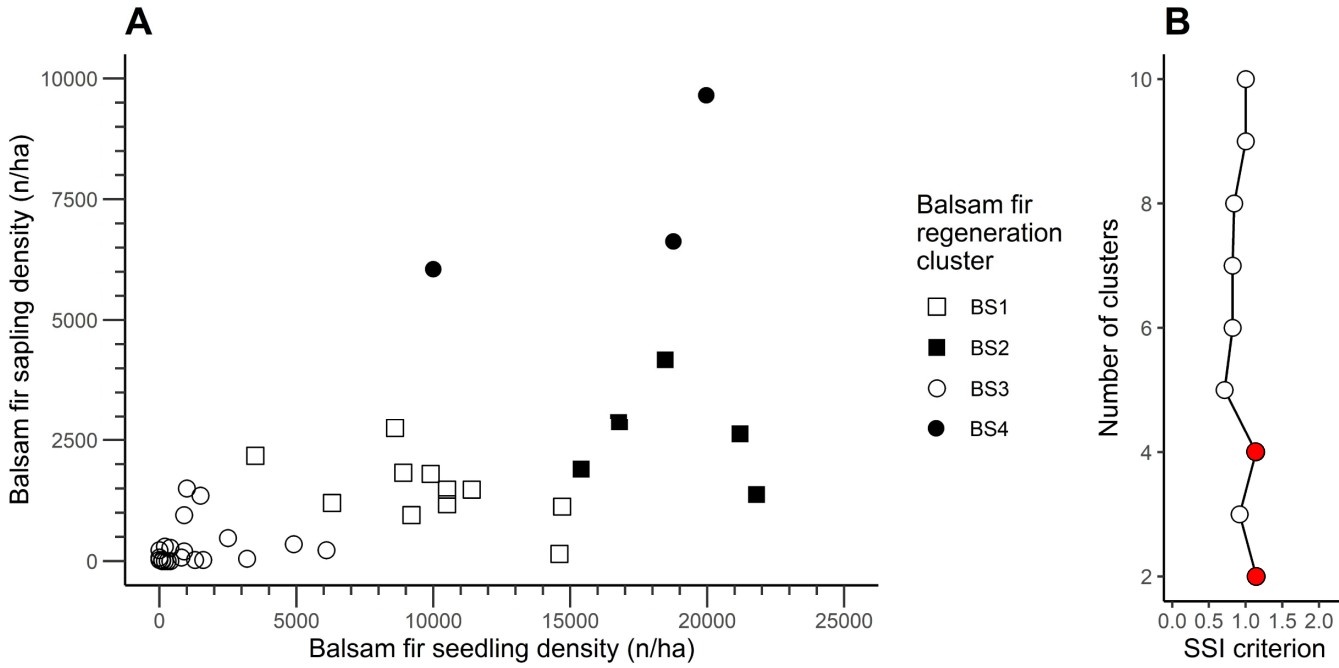

**Fig 4.** (A) Density of balsam fir seedlings and saplings at the 48 studied sites of the balsam fir regeneration portion of the study, grouped by balsam fir regeneration clusters. (B) Value of the SSI criterion according to the number of clusters for balsam fir, using k-means clustering. Filled red circles in (B) indicate the highest value of the SSI criterion.

**Table 3. Spearman correlation coefficients between regeneration attributes and structural and environmental attributes.**

| Category | Attribute | Black spruce ($n = 71$) | | Balsam firm ($n = 47$) | |
|---|---|---|---|---|---|
| | | Seedlings | Saplings | Seedlings | Saplings |
| Structure | Tree density (n/ha) | 0.18 | -0.09 | 0.14 | 0.10 |
| | Basal area (m$^2$/ha) | -0.09 | -0.49*** | 0.36* | 0.17 |
| | Balsam fir proportion (%) | -0.21 | -0.26* | 0.80*** | 0.86*** |
| | Gap fraction (%) | 0.34** | 0.51*** | -0.17 | -0.02 |
| | Weibull's shape parameter | 0.05 | 0.16 | -0.19 | -0.09 |
| | Coarse woody debris volume (m$^3$/ha) | -0.11 | -0.07 | 0.44** | 0.61*** |
| | Snag basal area (m$^2$/ha) | -0.21 | -0.21 | 0.55*** | 0.48*** |
| | Maximum height (m) | -0.12 | -0.31** | 0.39** | 0.33* |
| | Cohort basal area proportion | 0.32** | 0.24* | 0.08 | 0.07 |
| Abiotic | Minimum time since the last fire (years) | 0.43*** | 0.32** | -0.19 | -0.22 |
| | Slope (%) | -0.29* | -0.37** | 0.59*** | 0.56*** |
| | Organic horizon thickness (cm) | 0.41*** | 0.32** | -0.17 | -0.10 |

"*" indicates significance at $p < 0.05$

"**" at $p < 0.01$, and

"***" at $p < 0.001$.

## Structural and environmental attributes

Densities of black spruce seedlings and saplings both correlated positively with gap fraction, cohort basal area proportion, minimum time since the last fire, and thickness of the organic horizon; both correlated negatively with slope (Table 3; Fig 5). Black spruce seedling density correlated negatively with basal area, balsam fir proportion, and maximum height. Balsam fir seedling and sapling densities correlated positively with balsam fir proportion, coarse woody debris volume, snag basal area, maximum height, and slope (Fig 6). Balsam fir seedling density also correlated significantly with basal area. In general, correlation coefficients tended to be moderate even when significant; this was especially true for black spruce where only the correlation coefficient between sapling density and gap fraction exceeded 0.5. These moderate coefficient values indicate that no structural or environmental attributes presented a well-defined relationship with black spruce regeneration. We observed, however, elevated correlation coefficients ($\geq 0.5$) for balsam fir in relation to several structural and environmental attributes, including balsam fir proportion, slope, coarse woody debris volume (saplings only), and snag basal area (seedlings only).

Bootstrapped regressions explained 36.3% and 41.6% of the variance of black spruce seedling and sapling density, respectively (Table 4). For black spruce seedlings, the slope and the minimum time since the last fire presented the highest partitioned $R^2$ values (0.1). For black spruce saplings, however, gap fraction presented the highest partitioned $R^2$ values (0.12). All attributes generally had low confidence intervals, with the minimum values ranging near 0. These values suggest that none of the studied attributes could be linked clearly to black spruce regeneration by themselves, without invoking other attributes.

The bootstrapped regressions explained approximately 70% of balsam fir seedling and sapling density. The attribute of balsam fir proportion presented the highest partitioned $R^2$ values, both for balsam fir seedling density (0.4) and sapling density (0.2). In addition, the minimum values of the confidence interval were 0.2 for balsam fir seedling density and 0.1 for balsam fir sapling density. In contrast, the lowest confidence interval values for the other attributes were close to 0. All told, these patterns suggest that balsam fir proportion was the main attribute influencing balsam fir regeneration in old-growth stands.

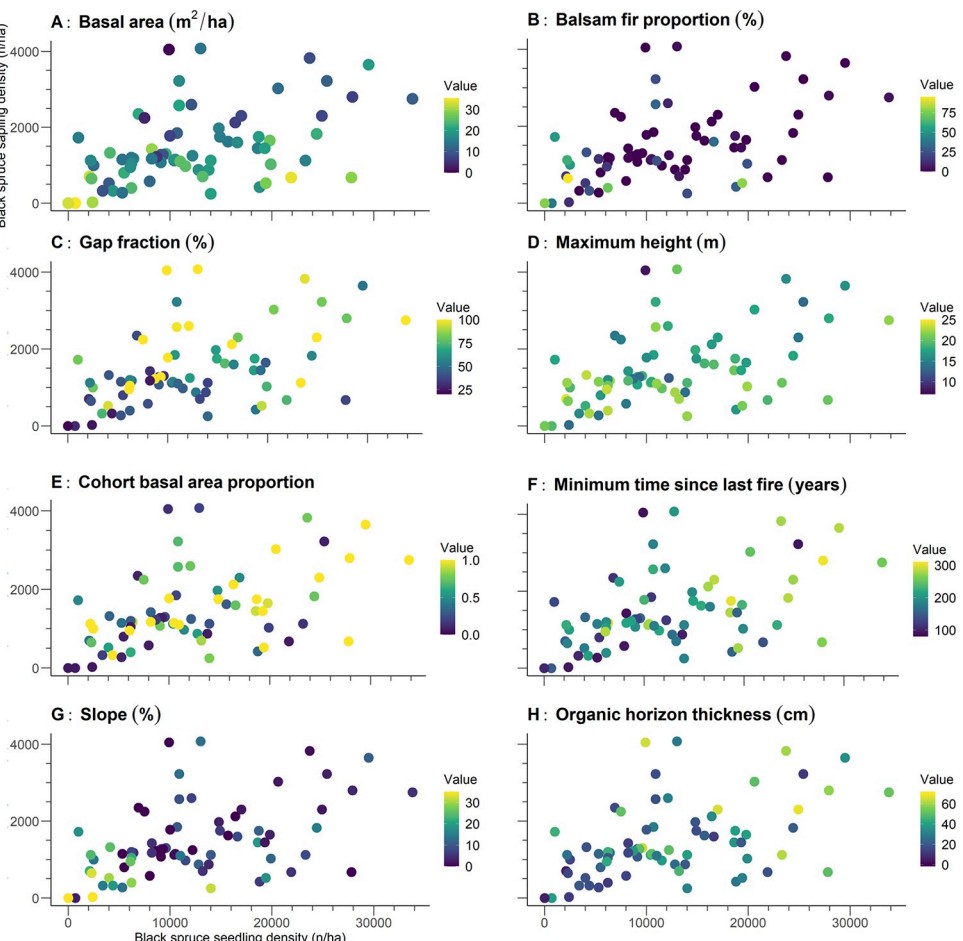

**Fig 5. Density of black spruce seedlings and saplings at the 71 studied sites.** The colors represent the gradients of values for the significant structural and environmental attributes, as determined by Spearman correlation tests.

Black spruce regeneration clusters differed significantly from each other for many attributes, including basal area, gap fraction, minimum time since the last fire, slope, and thickness of the organic horizon (Table 5). We identified marked differences between the study attributes and clusters; for example, basal area differed two-fold between cluster BS8 and cluster BS7, gap fraction values of cluster BS1 were more than double those of cluster BS5, the minimum time since the last fire varied between 146 (cluster BS1) to 249 years (cluster BS8), cluster BS8 had a 5× greater slope than that of cluster BS1 (4.0% versus 23.4%, respectively), and organic horizon thickness varied between 16.0 cm (cluster BS1) to 47.9 cm (cluster BS8). Overall, clusters BS1 and BS8 were the most distinct clusters; the other clusters fell along a gradient between this pair of clusters. Cluster BS1 grouped stands located on steeper sites, characterized by a thin organic horizon, a dense canopy, a high basal area, and relatively young trees. In contrast, cluster BS8 grouped stands having a gentle slope, a thick organic horizon, open canopy, low basal area, and older trees. The remaining clusters represented intermediate values between these two boundary clusters.

We noted significant differences between balsam fir regeneration clusters in terms of balsam fir proportion, coarse woody debris volume, snag basal area, and slope (Table 6). As with the black spruce regeneration clusters, two balsam fir regeneration clusters—clusters BF1 and BF4—represented opposite extremes along a gradient. Balsam fir proportion was almost 14×

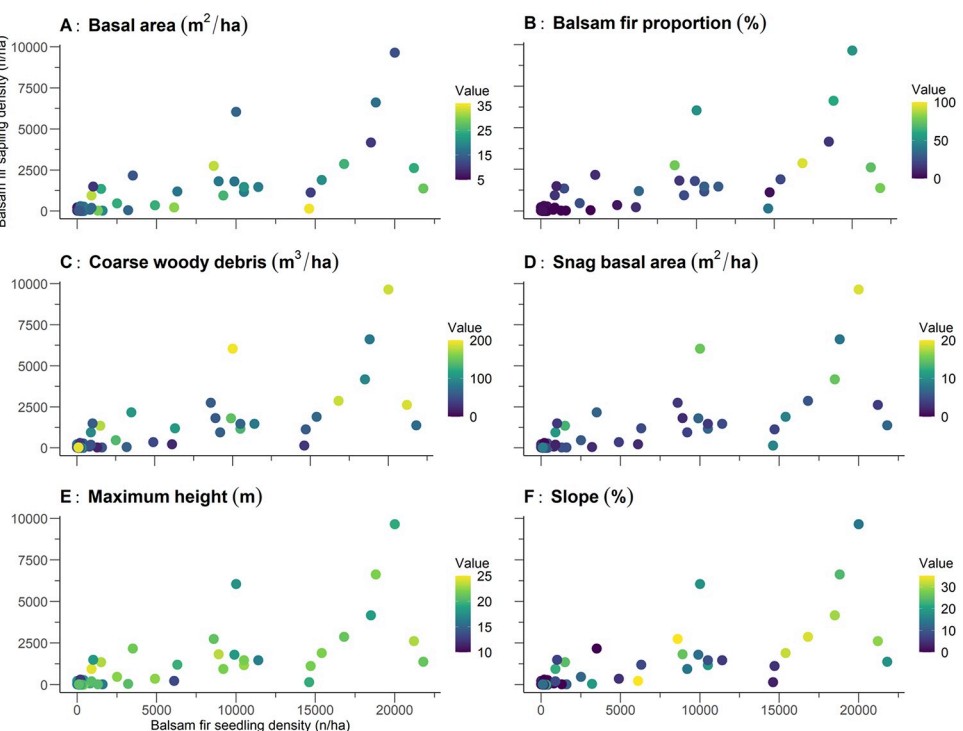

**Fig 6. Density of balsam fir seedlings and saplings at the 48 studied sites of the balsam fir regeneration portion of the study.** The colors represent the gradients of values for the various significant structural and environmental attributes, as determined by Spearman correlation tests.

higher in cluster BF4 (56.7%) than in cluster BF1 (4.12%). Coarse woody debris volume in cluster BF3 was more than double that of cluster BF1, at 61.6 and 155 m$^3 \cdot$ha$^{-1}$, respectively. Cluster BF4 contained a snag basal area that was more than triple that of cluster BF1 (14 versus 3.9 m$^2 \cdot$ha$^{-1}$, respectively). Slope in cluster BF4 (28.4%) was also 4× higher than that in cluster BF1 (8.14%). All told, cluster BF1 represented sites having a gentle slope and lower balsam fir proportion, a moderate coarse woody debris volume and moderate snag basal area. Cluster BF3, on the other hand, grouped sites marked by steeper slopes and higher values of balsam fir proportion, coarse woody debris volume, and snag basal area. As above, the remaining clusters fell between these two extreme clusters. Relative to the black spruce results, however, these balsam fir clusters differed much less from each other; for example, we observed no significant differences in coarse woody debris volume for clusters BF2, BF3, and BF4. This pattern implies that the structural differences within the balsam fir regeneration clusters were less noticeable than those observed in the black spruce stands.

## Discussion

Old-growth forests are critical habitats for biodiversity and ecosystem services. A better understanding of their functioning is therefore necessary for developing sustainable management strategies. The results of our study highlight that regeneration in boreal old-growth forests involves complex processes that cannot be summarized along a single linear chronosequence of forest succession or by using a limited number of structural attributes as proxies. In general, our analyses show that secondary disturbance regimes and topographic constraints (i.e., slope and drainage) represent the main drivers of balsam fir and black spruce regeneration in our study stands. Therefore, temporal (i.e., by way of the secondary disturbance regime) and

**Table 4. Mean values and 95% confidence intervals (C.I.) of the partitioned $R^2$ and total variance explained, of the bootstrapped regression models for the various regeneration attributes.**

| Explained variable | Explanatory variable | Partitioned $R^2$ | | Total variance explained (%) |
|---|---|---|---|---|
| | | Mean | 95% C.I. | |
| Black spruce seedling ($n = 71$) | Gap fraction | 0.03 | 0–0.14 | 36.39 |
| | Cohort basal area proportion | 0.04 | 0–0.13 | |
| | Slope | 0.1 | 0.02–0.21 | |
| | Min. time since the last fire | 0.1 | 0.02–0.22 | |
| | Org. horizon thickness | 0.07 | 0.01–0.17 | |
| Black spruce sapling ($n = 71$) | Basal area | 0.08 | 0.03–0.16 | 41.67 |
| | Balsam fir proportion | 0.03 | 0.01–0.08 | |
| | Gap fraction | 0.12 | 0.04–0.25 | |
| | Maximum height | 0.04 | 0–0.14 | |
| | Cohort basal area proportion | 0.01 | 0–0.09 | |
| | Min. time since the last fire | 0.02 | 0–0.11 | |
| | Slope | 0.05 | 0.01–0.13 | |
| | Org. horizon thickness | 0.02 | 0–0.1 | |
| Balsam fir seedling ($n = 47$) | Basal area | 0.02 | 0.01–0.09 | 72.8 |
| | Balsam fir proportion | 0.41 | 0.23–0.58 | |
| | Coarse woody debris volume | 0.05 | 0.02–0.13 | |
| | Snag basal area | 0.09 | 0.01–0.25 | |
| | Maximum height | 0.02 | 0–0.09 | |
| | Slope | 0.11 | 0.04–0.21 | |
| Balsam fir sapling ($n = 47$) | Balsam fir proportion | 0.2 | 0.1–0.4 | 66.97 |
| | Coarse woody debris volume | 0.1 | 0.02–0.22 | |
| | Snag basal area | 0.29 | 0.03–0.44 | |
| | Maximum height | 0.01 | 0–0.05 | |
| | Slope | 0.05 | 0.02–0.2 | |

spatial (i.e., local changes in topography) scales are two important factors for explaining the dynamics of tree regeneration in the boreal old-growth forests of eastern Canada.

## Dynamics of black spruce regeneration

The dynamics of black spruce regeneration in boreal old-growth forests involve particularly complex processes. We observed highly variable seedling and sapling densities within the study stands, and specific structural attributes defined each black spruce regeneration cluster. These observations may explain the generally moderate Spearman correlation coefficients and the absence of any determinant attribute that could fully explain variations in black spruce regeneration. These results show that black spruce regeneration density depends on multiple and interrelated factors [10,15,64]. Overall, two clusters representing two extreme environmental conditions at the study area scale, i.e., BS1 and BS8, best define black spruce regeneration patterns. BS1 contained a thin organic horizon, a steep slope, and a low gap fraction. It was also the cluster with the lowest black spruce seedling and sapling densities. This pattern matches prior observations in the study area where the abundance of black spruce decreases progressively as slope increases, eventually being replaced by balsam fir and northern hardwoods [40,45]. Competition with balsam fir could explain the limited regeneration of black spruce on these steepest sites as their ecological niches overlap [65]. However, black spruce was the dominant species in most stands associated with this cluster. This implies that black

**Table 5. Mean and standard deviation of the structural and environmental attributes for black spruce regeneration clusters (BS).**

| Cluster | BS1 ($n = 10$) | BS2 ($n = 11$) | BS3 ($n = 17$) | BS4 ($n = 6$) | BS5 ($n = 3$) | BS6 ($n = 11$) | BS7 ($n = 6$) | BS8 ($n = 7$) |
|---|---|---|---|---|---|---|---|---|
| Tree density (n/ha)[†] | 790.00 ± 332.00 | 950.00 ± 392.00 | 899.00 ± 283.00 | 925.00 ± 569.00 | 600.00 ± 563.00 | 1 068.00 ± 382.00 | 1 162.00 ± 423.00 | 832 ± 399 |
| Basal area (m²/ha)[†] | 23.20 ± 9.93 ab | 18.10 ± 5.16 abc | 17.20 ± 5.83 bc | 12.40 ± 7.47 cd | 10.40 ± 8.84 cd | 16.60 ± 5.85 bcd | 25.10 ± 7.01 a | 11.10 ± 4.32 d |
| Balsam fir proportion (%)[‡] | 35.20 ± 34.30 | 20.00 ± 23.90 | 3.60 ± 7.64 | 6.91 ± 12.80 | 8.79 ± 12.40 | 3.99 ± 10.10 | 22.30 ± 30.70 | 1.16 ± 1.52 |
| Gap fraction (%)[‡] | 42.70 ± 23.70 c | 61.60 ± 25.00 ab | 49.80 ± 21.10 bc | 83.40 ± 26.40 a | 85.30 ± 25.40 a | 66.60 ± 15.50 ab | 71.30 ± 23.80 ab | 84.80 ± 17.10 a |
| Weibull's shape parameter[‡] | 1.11 ± 0.68 | 1.07 ± 0.46 | 1.03 ± 0.43 | 1.09 ± 0.17 | 1.06 ± 0.22 | 1.05 ± 0.20 | 0.80 ± 0.45 | 0.98 ± 0.16 |
| Coarse woody debris volume (m³/ha)[‡] | 82.30 ± 69.10 | 92.00 ± 69.50 | 33.20 ± 22.70 | 27.60 ± 29.20 | 113.00 ± 101.00 | 51.00 ± 34.10 | 60.70 ± 48.50 | 41.90 ± 25.40 |
| Snag basal area (m²/ha)[‡] | 5.80 ± 4.68 | 7.27 ± 6.00 | 3.57 ± 2.36 | 2.71 ± 1.17 | 5.17 ± 4.59 | 2.72 ± 1.52 | 4.42 ± 1.97 | 2.93 ± 1.54 |
| Maximum height (m)[‡] | 19.30 ± 3.88 | 20.00 ± 2.77 | 18.10 ± 3.37 | 17.10 ± 2.78 | 15.60 ± 6.78 | 18.10 ± 1.22 | 20.00 ± 1.98 | 16.80 ± 2.78 |
| Cohort basal area proportion[‡] | 0.37 ± 0.33 | 0.53 ± 0.37 | 0.48 ± 0.35 | 0.58 ± 0.35 | 0.34 ± 0.29 | 0.78 ± 0.25 | 0.45 ± 0.42 | 0.83 ± 0.34 |
| Minimum time since the last fire (years)[‡] | 146.00 ± 45.60 c | 190.00 ± 65.40 bc | 179.00 ± 52.90 bc | 181.00 ± 53.00 bc | 159.00 ± 61.60 bc | 239.00 ± 49.70 a | 209.00 ± 56.40 ab | 249.00 ± 71.60 a |
| Slope (%)[‡] | 23.40 ± 10.80 a | 13.20 ± 10.30 b | 6.35 ± 8.03 c | 4.33 ± 4.84 c | 8.67 ± 7.51 bc | 6.27 ± 6.33 bc | 8.17 ± 6.21 bc | 4.00 ± 3.37 c |
| Organic horizon thickness (cm)[‡] | 16.00 ± 9.73 c | 27.10 ± 11.80 b | 35.10 ± 14.30 ab | 27.70 ± 13.00 b | 37.00 ± 25.00 ab | 29.90 ± 15.40 b | 37.20 ± 16.00 ab | 47.90 ± 18.80 a |

Different letters indicate significant differences at $p < 0.05$, following a > b > c > d. BS: black spruce; BF: balsam fir.

spruce trees were abundant enough to produce a high density of layers in these stands. Thus, it is also possible that environmental conditions within these stands were unfavorable to layering, thereby accentuating the competitiveness of balsam fir. For example, Drobyshev et al. [66] highlighted that a thin organic horizon reduced the survival of black spruce layers because of insufficient moisture, especially in the summer. This role of thin organic horizons may thus reinforce the competitiveness of balsam fir relative to black spruce in steep-sloped sites. In contrast, cluster BS8 contained a gentle slope, a thick organic horizon, a high gap fraction, and a small basal area. These characteristics typify stands undergoing paludification—the accumulation of soil organic matter due to insufficient drainage resulting in a decreased stand productivity [67,68]. Paludification inhibits tree growth but not black spruce regeneration. As a result, black spruce sapling and seedling densities are commonly quite dense in paludified black spruce stands, but these saplings and seedlings are unable to close the gaps caused by overstory tree death [29]. Paludification, however, is a process limited to specific conditions, i.e., poor drainage and low temperatures; this process is not observed within well or moderately well drained soils, i.e., stands having at least a minimum of slope [69–71], thereby highlighting the particularity of this cluster.

The remaining black spruce regeneration clusters differed to a much lower degree; these clusters contained similar slopes and organic horizon thicknesses, implying similar environmental conditions among these clusters. We observed a significant difference between clusters in relation to the minimum time since the last fire; however, this value generally exceeded 150 years, i.e., the threshold beyond which tree age becomes a poor indicator of stand age in boreal forests [72,73]. The longevity of both black spruce and balsam fir is relatively limited and rarely exceeds 200 years, and mean tree age in old-growth stands in the study area is approximately

**Table 6. Mean ± standard deviation of structural and environmental attributes for balsam fir regeneration clusters (BF).**

| Cluster | BF1 (*n* = 28) | BF2 (*n* = 11) | BF3 (*n* = 3) | BF4 (*n* = 5) |
|---|---|---|---|---|
| Tree density (n/ha)‡ | 880.00 ± 332.00 | 927.00 ± 277.00 | 892.00 ± 104.00 | 810.00 ± 326.00 |
| Basal area (m²/ha)‡ | 17.50 ± 7.27 | 20.80 ± 7.64 | 14.90 ± 1.92 | 21.70 ± 6.94 |
| Balsam fir proportion (%)‡ | 4.12 ± 6.95 b | 28.80 ± 19.10 a | 55.00 ± 5.10 a | 56.70 ± 35.40 a |
| Gap fraction (%)‡ | 64.10 ± 26.00 | 57.10 ± 29.80 | 72.70 ± 14.80 | 64.00 ± 27.10 |
| Weibull's shape parameter‡ | 0.87 ± 0.29 | 1.15 ± 0.62 | 0.88 ± 0.12 | 0.81 ± 0.13 |
| Coarse woody debris volume (m³/ha)‡ | 61.60 ± 47.00 b | 84.00 ± 35.60 a | 155.00 ± 62.90 a | 121.00 ± 60.00 a |
| Snag basal area (m²/ha)‡ | 3.90 ± 3.05 c | 5.09 ± 2.00 bc | 14.00 ± 5.58 a | 7.97 ± 4.40 ab |
| Maximum height (m)† | 18.90 ± 3.05 | 20.70 ± 2.06 | 19.70 ± 2.23 | 21.20 ± 1.75 |
| Cohort basal area proportion‡ | 0.60 ± 0.35 | 0.63 ± 0.36 | 0.82 ± 0.30 | 0.74 ± 0.16 |
| Minimum time since the last fire (years)† | 213.00 ± 66.50 | 193.00 ± 50.60 | 188.00 ± 50.10 | 204.00 ± 41.40 |
| Slope (%)‡ | 8.14 ± 9.11 c | 12.50 ± 10.5 bc | 18.70 ± 5.03 ab | 28.40 ± 6.02 a |
| Organic horizon thickness (cm)‡ | 31.60 ± 16.00 | 26.10 ± 14.00 | 29.00 ± 15.10 | 21.60 ± 9.29 |

Letters indicate significant differences at *p* < 0.05, following a > b > c.

150 years [30,58,74]. Differences in the minimum time since the last fire between clusters BS2 to BS7 are therefore not necessarily indicative of a distinction in terms of forest succession. On the contrary, a lower minimum time since the last fire value may indicate the death of older, and therefore more vulnerable, trees as a result of a secondary disturbance, such as windthrow or spruce budworm outbreak [37,75,76]. For this reason, Kneeshaw and Gauthier [52] proposed the cohort basal area proportion as a more reliable alternative for reconstructing forest succession without depending on the age of the oldest trees.

Yet, black spruce regeneration clusters presented no significant differences in their cohort basal area proportion; therefore, differences between clusters did not necessarily result from succession toward an old-growth stage. The clusters were, however, defined by different basal areas and gap fractions. As we observed no differences in tree density between the clusters, it is therefore likely that these changes in basal area and gap fraction indicate variations in tree size and canopy density. However, we saw no differences in snag basal area or coarse woody debris volume, both of which can indicate secondary disturbances. Nonetheless, individual tree size is generally low in black spruce–dominated stands [40,47], and deadwood can be buried in the organic horizon well before its decay [77]. This particularity of black spruce–dominated old-growth stands may hence explain the absence of any significant results for deadwood-related attributes. Overall, these results are likely to indicate the influence of secondary disturbances (e.g., windthrow or spruce budworm outbreak), either in a context of forest succession (i.e., canopy breakup) or gap dynamics, on black spruce regeneration in old-growth forests.

From the information provided by the basal area and the gap fraction of the clusters BS2 to BS7, we can hypothesize a general pathway of black spruce regeneration in old-growth forests under a dominance of gap dynamics (Fig 7A). It is likely, however, that the regeneration of black spruce will vary greatly depending on the nature, the severity, and the temporality of a disturbance. For example, the death of a single tree due to senescence has less effect on regeneration and overall stand structure than a severe spruce budworm outbreak. Therefore, this pathway may represent the overall boundaries of the characteristics of black spruce regeneration in old-growth forests. As a starting point to these potential regeneration pathways for black spruce under a secondary disturbance regime, cluster BS7 grouped dense old-growth forest stands found on gentle to medium slopes (0–7% and 8–24%, respectively). The stands in

## A: Black spruce regeneration

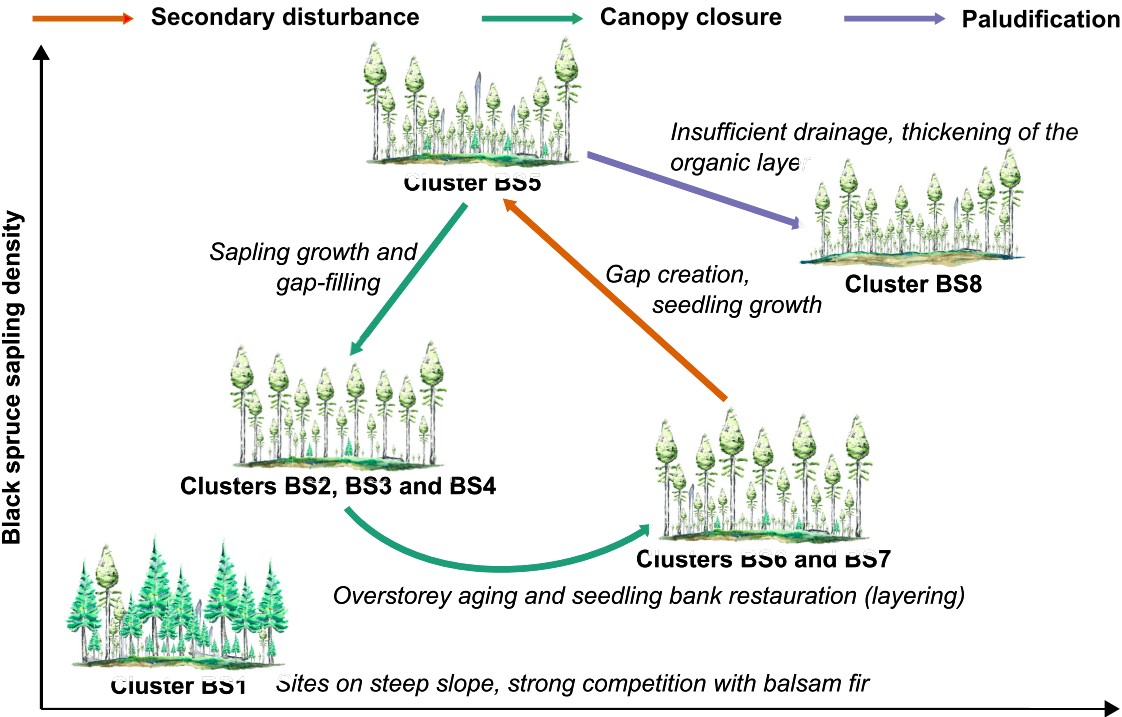

## B: Balsam fir regeneration

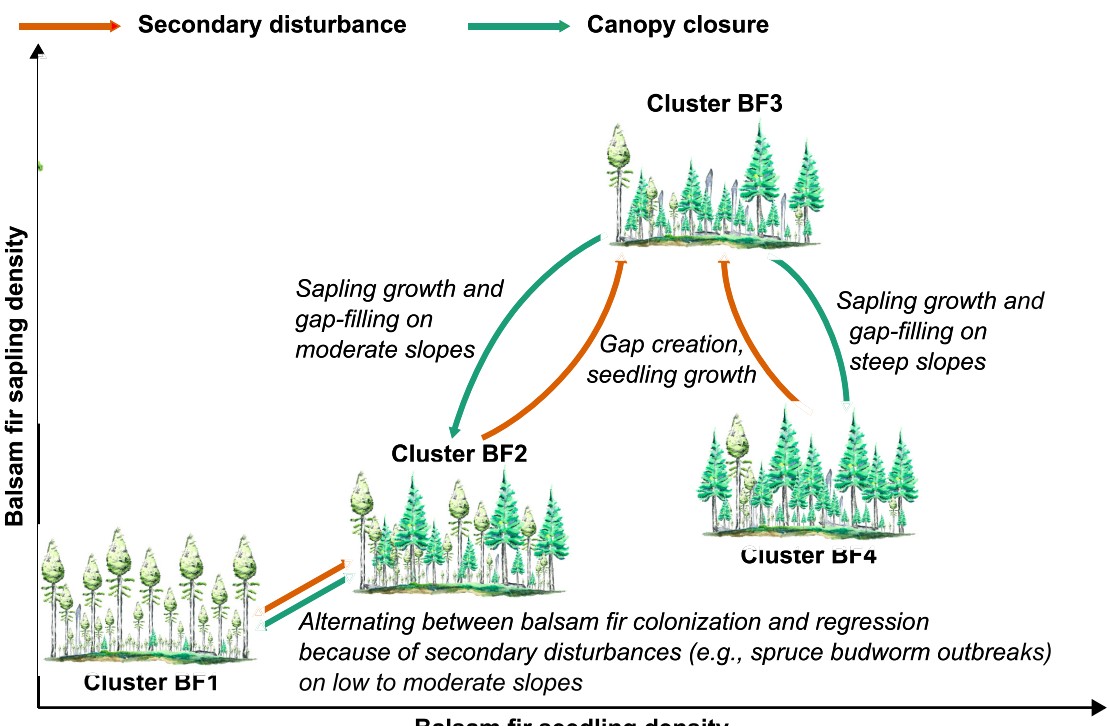

**Fig 7.** Dynamics of (A) black spruce and (B) balsam fir regeneration based on secondary disturbance regime and topography as derived from the identified regeneration clusters. Water paintings published under a CC BY license, with permission from Valentina Buttò, original copyright 2019.

this cluster contained a moderate gap fraction and a large basal area, i.e., stands that have most probably neither been recently nor significantly disturbed. Indeed, because of their narrow canopy, even dense old-growth black spruce stands can be characterized by a relatively high gap fraction [41]. At this cluster's successional stage, a low black spruce sapling density and high seedling density indicated a dense understory waiting for a canopy opening. Here, we can hypothesize that overstory trees eventually die to create gaps and reduce the stand basal area. Black spruce regeneration individuals, including layers, are efficient gap-fillers [26,30,33], and most seedlings benefit from these openings, thereby increasing sapling density (i.e., a shift from cluster BS6 to BS5). Saplings eventually attain the overstory and progressively close the canopy. This growth causes a significant decrease in sapling density. From this point, the pathways may diverge, depending on stand topography: gentle slopes (clusters BS4 and BS3, sapling growth and canopy closure, respectively) and moderate slopes (cluster BS2, sapling growth and canopy closure). Finally, we can expect that canopy closing leads to an increased stand basal area, i.e., clusters BS2 and BS3 shift toward cluster BS7, reinitiating the cycle.

Our results also highlighted that the sapling density of black spruce is commonly low in the denser old-growth stands, even where the seedling density is very high. A relatively low abundance of saplings or small-diameter merchantable stems has been observed in old-growth forests dominated by black spruce [30,78,79]. This pattern is, however, counterintuitive, as old-growth forests are generally expected to have a strongly stratified vertical structure and a high sapling or small tree density [8,80,81]. Lower light availability in the understory is unlikely to explain this result. Indeed, the canopies of the study stands were rarely dense, and black spruce can grow quite easily in conditions of low light availability [82–84]. Similarly, marked competition between black spruce regeneration and shrubs is generally observed after clearcutting but not after secondary disturbances [64,85]. However, little is known about the vegetative relationship between black spruce layers and the mother trees. Studies focused on layer growth generally observe black spruce stands post-clearcutting or after a severe spruce budworm outbreak [33,86,87]. In contrast, in situ research on planted seedlings has assessed the sensitivity of black spruce regeneration to competition or light availability [82–84]. It is possible that black spruce layers likely remain under hormonal control with the process of apical dominance inhibiting their growth (lateral growth) [88–90] until the mother tree dies of senescence or a disturbance occurs. This link between layers and mother trees may explain the following patterns of black spruce regeneration in old-growth forests: (1) development of a dense seedling bank of layers under a closed canopy; (2) seedling growth once the overstory is disturbed and the link between layers and mother trees is broken, thereby decreasing seedling density and increasing sapling density; (3) progressive canopy closure, implying a decrease in sapling density as saplings become merchantable trees; and (4) new production of layers by black spruce, leading to a return to phase 1. Further research is required to better understand to what extent the vegetative nature of black spruce regeneration in old-growth forests may influence their structure and dynamics.

### Balsam fir regeneration dynamics

Disentangling balsam fir regeneration dynamics in our study stands presented a greater challenge than for black spruce dynamics because balsam fir regeneration was absent from 24 plots and sparse for the 28 sites belonging to cluster BF1. Several factors may explain the scarcity of

balsam fir regeneration in most of the studied stands, including the soils being too wet or the stands having a limited seed bank. In sites characterized by relatively poor drainage, very wet and cold soils inhibit balsam fir seed germination and favor black spruce layering [91,92]. In the study region, the fire cycle is shorter in the valley bottoms than on the hilltops [45], probably related to later snowmelt at higher elevations. Balsam fir is not a fire-adapted species, and this tree requires decades, if not centuries, to recolonize a burned area [93]. Moreover, the dispersal of balsam fir seeds is relatively limited, and its occurrence requires proximal seed-trees [15,39] as evidenced by the strong correlation observed between the proportion of balsam fir and the balsam fir regeneration density. Shorter fire cycles in the valley bottoms may thus inhibit the colonization of balsam fir in these areas of the study territory. Nonetheless, the absence of balsam fir in boreal old-growth stands is common in eastern Canada [19,20,40] because of all the previously explained factors; thus, sampling bias does not account for the results in our study.

We observed no significant difference between the balsam fir regeneration clusters in terms of the minimum time since the last fire and the cohort basal area proportion. As with the black spruce clusters, all balsam fir clusters represented the old-growth successional stage. Previous research of balsam fir regeneration dynamics in the boreal forests of eastern Canada focused on stands at the beginning of the transition toward the old-growth stage (e.g., [27,94,95]). Our results underscore that once the old-growth stage is attained, and if seed-trees are present nearby, the existing seed bank is sufficient to provide continuous regeneration of balsam fir [28,96]. Moreover, we observed significantly different slopes between the clusters, highlighting the importance of topography in explaining balsam fir stand dynamics [40,65]. These results imply that, as with black spruce, secondary disturbance dynamics and topographic constraints drive balsam fir regeneration in the old-growth forests of eastern Canada. Indeed, balsam fir is driven by the same disturbances as black spruce (i.e., spruce budworm outbreak, windthrow, and root rot) [75,97]; however, balsam fir is generally more sensitive to these disturbances than black spruce [35–37]. In the case of windthrow, the higher sensitivity of balsam fir may be reinforced by the greater abundance of this tree species on hilly areas, which are more vulnerable to strong winds [98]. For these reasons, we observe, in general, evidence of a more severe secondary disturbance regime in the old-growth forests where balsam fir forms a major part of the canopy [20].

For sites located on gentle slopes (0–8%), we observed two different balsam fir regeneration clusters. One cluster represented sites where balsam fir was almost absent from the canopy (BF1), whereas the other cluster represented stands where balsam fir accounted for approximately 30% of the basal area (BF2). As a result, there was almost no balsam fir regeneration in BF1, whereas seedling and sapling densities were of moderate levels in BF2. Coarse woody debris volume was, however, higher in BF2 than BF1, suggesting more recent disturbances (Fig 7B). This involves a dynamic where boreal old-growth species composition switches between a pure black spruce stand and a mixed black spruce and balsam fir stand, possibly with the presence of white birch at a very low abundance [27,28]. This type of dynamic is consistent with previous observations [28,40]. Balsam fir is a competitive species that can quickly reach the upper canopy following a secondary disturbance [28,31]. Balsam fir is also very sensitive to spruce budworm outbreaks, the main secondary disturbance agent in eastern Canadian boreal forests [37,99,100]. Outbreaks of this insect heighten balsam fir mortality because spruce budworm larvae emergence is well synchronized with balsam fir budburst. In contrast, black spruce mortality during spruce budworm outbreaks is relatively low because black spruce budburst and larval emergence are poorly synchronized [101]. The most severe budworm outbreaks cause significant mortality of the regeneration, in particular that of balsam fir [38,102,103]. For this reason, it is expected that spruce budworm outbreaks may significantly

reduce the portion of balsam fir in the disturbed stands [20,21,28]. As a consequence, mixed black spruce–balsam fir stands may shift to monospecific black spruce stands. Balsam fir may, however, progressively recolonize these stands over time. Hence, we propose that clusters BS1 and BS2 represent this process of alternating between monospecific and mixed stands in environments where black spruce remains competitive with balsam fir.

We observed no differences between the balsam fir regeneration clusters BF2 and BF3 in terms of coarse woody debris volume and the proportion of balsam fir; this pattern represents dynamics in sites of moderate slope (i.e., 9–28%). The snag basal area, however, was significantly higher in BF3. Relative to black spruce, balsam fir is also more vulnerable to windthrow and fungal rot [35,36]. The presence of an important coarse woody debris volume in stands characterized by an elevated balsam fir proportion in the canopy is therefore consistent with balsam fir ecology. A higher snag basal area can, however, also indicate a relatively recent disturbance, as black spruce and balsam fir snags often fall in the twenty years following a tree death [104]. Thus, cluster BF3 may group recently disturbed stands marked by a dynamic balsam fir regeneration that quickly fills the canopy [27,28,31]. Hence, we suggest a pathway where once the canopy is closed, the stand structure of BF3 shifts to BF2, defined by a dense seedling bank.

Finally, BF3 and BF4 grouped stands found on steep slopes (>28%), although significant structural differences existed between the two clusters. This result may reflect the low number of sites sampled for both clusters (3 and 5 sites, respectively). However, it is also probable that they represented a balsam fir regeneration dynamic similar to that observed on moderate slopes, with BF3 grouping recently disturbed stands and the BF4 grouping the resilient stands. On intermediate slopes, black spruce regeneration continued to compete with balsam fir, thereby explaining the intermediate balsam fir seedling density in BF2. On steep slopes, however, balsam fir dominated the canopy. It is therefore likely that these stands were driven by regular small- and moderate-scale disturbances [26], resulting in recurrent deadwood inputs and active regeneration/mortality phases.

## Synchronic reconstruction of the regeneration dynamics of old-growth forests: Limits and alternatives

Our study of coniferous regeneration dynamics in old-growth forests relied on a synchronic approach. We therefore determined successional processes and disturbance dynamics using indirect indicators, including gap fraction, deadwood volume, and stand age structure. This approach therefore has inherent limits. First, the dynamics are reconstructed on the basis of an interpretation founded on current ecological knowledge and not on direct observation. As well, it is likely that the studied stands are not fully comparable, for instance in terms of environmental conditions or disturbance history. As an example, the thickness of the residual organic layer after a stand-replacing fire will markedly influence post-fire stand regeneration and dynamics [105,106]. Thus, stands of similar age since the last fire and having similar environmental conditions can nonetheless be defined by significantly different structures or tree species compositions depending on the characteristics of the last stand-replacing disturbance. Yet, diachronic surveys are hard to implement in old-growth forests owing to the extensive time required to reach this successional stage after stand initiation. At a shorter time scale, even the complete filling of canopy gaps can require several years, if not decades, because of the relatively low growth rate of boreal tree species [72,107]. For these reasons, most research on forest dynamics within the study territory rely, at least partially, on a synchronic approach (e.g., [27,28,47]).

The greater availability of aerial and satellite imagery over longer temporal scales could offer an opportunity to better understand forest dynamics over time [108,109]. The spatial

resolution of these images, however, may not be adapted to fine-scale processes, such as regeneration under gap dynamics.

Overall, the study of long-term regeneration dynamics under a secondary disturbance regime remains limited in old-growth boreal forests. Similarly, the internal complexity of old-growth boreal forests remains poorly understood, as most studies on this topic rely on chronosequences defined by tree age or time since the last fire [47,79,110]. Despite its limitations, the synchronic approach, as used in this study, provides a conceptual framework able to provide a basis for future research. Original and long-term experimental designs are required to confirm the validity of some of the regeneration patterns suggested in this study.

## Conclusion

This study determined how secondary disturbance regimes and topographic constraints explain the dynamics of black spruce and balsam fir regeneration in old-growth forests. Our study offers an alternative perspective to standard models of forest dynamics, which generally rely on chronosequences. Old-growth forest dynamics are complex processes involving numerous ecological factors, which interact at multiple temporal and spatial scales. The progressive changes in forest structure, composition, and regeneration may follow various pathways that result, not only from stand aging, but also from interactions between stand structure, stand history, characteristics of the disturbance agents, and local environmental conditions.

Second, this study provides a better acknowledgment of regeneration dynamics in the boreal old-growth forests of eastern Canada. Disturbance dynamics in these ecosystems are, however, defined by disturbances that vary in terms of type, frequency, and severity [26,111]. Thus, our results highlight the potential pathways of regeneration dynamics in old-growth forests, and further research is required to determine how these trends may change depending on disturbance characteristics.

Third, sustainable forest management aims to develop new silvicultural treatments to minimize differences between natural and managed stands. For this, partial cuttings offer a promising solution to adapt forestry practices to act in a similar manner as secondary disturbance regimes. These treatments, however, were previously developed in Europe and must be carefully adapted to the environmental and functional characteristics of Canadian boreal forests [112–115]. The results of our study provide new guidelines for a forest management approach that brings the regeneration dynamics within managed stands closer to those of boreal old-growth forests.

## Acknowledgments

We thank Audrey Bédard, Jean-Guy Girard, Émilie Chouinard, Anne-Élizabeth Harvey, Aurélie Cuvelière, Évelyn Beliën, and Angelo Fierravanti for their precious help during field sampling. Yan Boucher and Pierre Grondin from Quebec's Ministry of Forests, Fauna, and Parks (MFFP) provided the data collected from the study territory. We also thank Valentina Buttò for her water paintings, used in Fig 4 of this paper.

## Author Contributions

**Conceptualization:** Maxence Martin, Miguel Montoro Girona.

**Data curation:** Maxence Martin.

**Formal analysis:** Maxence Martin.

**Funding acquisition:** Miguel Montoro Girona, Hubert Morin.

**Investigation:** Maxence Martin.

**Methodology:** Maxence Martin.

**Project administration:** Maxence Martin.

**Resources:** Hubert Morin.

**Software:** Maxence Martin.

**Supervision:** Miguel Montoro Girona, Hubert Morin.

**Validation:** Maxence Martin, Miguel Montoro Girona, Hubert Morin.

**Visualization:** Maxence Martin.

**Writing – original draft:** Maxence Martin, Miguel Montoro Girona, Hubert Morin.

**Writing – review & editing:** Maxence Martin.

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
