## [Decision Letter · Decision Letter 0]

10 May 2020

PONE-D-20-04997

Driving factors of conifer regeneration dynamics in eastern Canadian boreal old-growth forests

PLOS ONE

Dear Dr. Martin,

Thank you for submitting your manuscript to PLOS ONE. After careful consideration, we feel that it has merit but does not fully meet PLOS ONE’s publication criteria as it currently stands. Therefore, we invite you to submit a revised version of the manuscript that addresses the points raised during the review process.

ACADEMIC EDITOR: I agree with the reviewer that the manuscript is generally a well written paper,presenting some important new findings about boreal forest dynamics.As indicated by the reviewer,you should make clarifications in terminology and methods,and make the discussions be more closely relevant to the results.

We would appreciate receiving your revised manuscript by Jun 24 2020 11:59PM. To enhance the reproducibility of your results, we recommend that if applicable you deposit your laboratory protocols in protocols.io, where a protocol can be assigned its own identifier (DOI) such that it can be cited independently in the future. For instructions see: http://journals.plos.org/plosone/s/submission-guidelines#loc-laboratory-protocols

We look forward to receiving your revised manuscript.

Kind regards,

RunGuo Zang

Academic Editor

PLOS ONE

Additional Editor Comments (if provided):

Please consider the concerns of the referee,make improvement accordingly.

3. We note that Figure 1 in your submission contains map images which may be copyrighted. All PLOS content is published under the Creative Commons Attribution License (CC BY 4.0), which means that the manuscript, images, and Supporting Information files will be freely available online, and any third party is permitted to access, download, copy, distribute, and use these materials in any way, even commercially, with proper attribution. For these reasons, we cannot publish previously copyrighted maps or satellite images created using proprietary data, such as Google software (Google Maps, Street View, and Earth). For more information, see our copyright guidelines: http://journals.plos.org/plosone/s/licenses-and-copyright.

Reviewers' comments:

Reviewer's Responses to Questions

**Comments to the Author**

1. Is the manuscript technically sound, and do the data support the conclusions?

Reviewer #1: Partly

2. Has the statistical analysis been performed appropriately and rigorously? 

Reviewer #1: Yes

3. Have the authors made all data underlying the findings in their manuscript fully available?

Reviewer #1: No

4. Is the manuscript presented in an intelligible fashion and written in standard English?

Reviewer #1: Yes

5. Review Comments to the Author

Reviewer #1: SUMMARY

In this study, Martin et al. investigate the patterns and drivers of forest regeneration dynamics in boreal old-growth forests of Canada. Specifically, they assessed stand structural and environmental characteristics of 71 systematically selected forest stands and relate these to differences in black spruce (Picea mariana (L.) Mill.) and balsam fir (Balsam fir (L.) Mill.) regeneration attributes measured by seedling and sapling densities in the years 2015 & 2016. They find that forest regeneration dynamics differ among black spruce and balsam fir species. Nevertheless, in both cases, secondary disturbance and topography were the main drivers explaining regeneration attributes, whereas successional stage and primary disturbance are likely less important. These findings help to shed light on old-growth forest regeneration dynamics and contribute to improving forest management and restoration practices.

GENERAL COMMENTS

The study addresses an important topic, which is not only of high interest to the scientific community, but also to the forest managers and conservationists, as it could guide future silvicultural and restoration practices. In general, the manuscript is well written and methods well described. I like the comparative approach including k-means grouping and a final schematic description of the inferred, general black spruce and balsam fir regeneration processes. Nevertheless, there are some minor shortcomings that need to be addressed before the article can be published.

1) Concerning “secondary disturbance”: It is not entirely clear to me which kind of disturbance is meant by this notion; could you please more clearly state which processes you refer to in the Introduction? I.e. do you mean pathogen (budworm) outbreaks, windthrow, root rot, fire, and logging? Or do you consider “fire” and “logging” as a primary disturbances?

Could explicitly explain in the Methods section how exactly you inferred the secondary disturbance (gap fraction and amount of deadwood,..?) and why these measures are appropriate proxies for secondary disturbance? Finally, it would be great if you could shortly discuss in the Discussion the advantages & disadvantages of approximating secondary disturbances with gap fraction and amount of deadwood?

Example that needs clarification:

Line 144 ff.: Hence, is fire no secondary disturbance? Is logging considered a secondary disturbance?

Line 381 ff.: could you please elaborate which types of secondary disturbances you are referring to here?

Line 470: Please elaborate on the kind of disturbance of balsam fir, which might be different from black spruce?

2) Concerning the notion of “topography”: To approximate “topography” you mainly measured “slope”. To improve the clarity throughout the manuscript, I would add the notion of slope in parentheses after “topography” or entirely replace the notion of “topography” with the notion of “slope”. Also, please explain why you chose “slope” as an important factor representative of “topography” and why you did not test effects of other topographic parameters such as altitude?

3) Methods: Some important aspects of the methods are not clear:

Lines 154 ff.: When did you take measurements? During summers or the whole years of 2015/2016? Once per plot in these 2 years? Or once per plot per year?

Line 154: Stratified random sampling approach: Did you select from a larger set of possible forest stands? If yes from where (how many sites in total were available)? Also, 71 is not a multiple of 6; therefore not all environmental types are represented with the same amount of replicates? Could you indicate the number and distribution of selected forest stands that correspond to the environmental types? (e.g. you could add 6 different colours instead of the red colour for the dots in Figure 1)?

Line 161: What is the “productive forest”? Did you not select from “unproductive forest”? I do not understand the meaning of this sentence.

Line 190: Could you explain in more detail how you measured gap length? Did you use hemispheric photography to do this?

-Somewhere in this section it would be nice to have a clearer/more structured description that you approximated “topography” by measuring slope using a clinometer, and that you approximated “secondary disturbance” by assessing “gap fraction” and “woody debris”.

Line 210: Did (40) measure the structural attributes in the same time period as the attributes presented in this study?

4) Analyses/Results:

Line 257: Could you elaborate a bit more about the significance of the SSI? I.e. what are high/low values? Is a grouping with a SSI of 2.23 much better than a grouping with SSI of 1.5 or only slightly better?

Line 303 & 322 ff.: This Tables 3, 4 & 5 somehow show results that are not entirely consistent with your main message that “secondary disturbance” and “topography (i.e. slope?)” were the main drivers of regeneration in black spruce and balsam fir:

-e.g. black spruce seedling and sapling densities significantly correlate to cohort basal area proportion & minimum time since last fire (both measures of “succession” as far as I understand[?]) & also with soil organic layer depth. This is also reflected in cluster differences of Table 4. Unless I misunderstood your main message, I think it should be mentioned in the Discussion that in case of black spruce, successional stage and soil organic layer are also important drivers of regeneration dynamics.

-e.g., balsam fir significantly relates to woody debris and snag basal area and not to gap fraction, whereas black spruce relates mainly to gap fraction. Hence the two species are influenced by different secondary disturbance drivers: Could you elaborate on this more in the Discussion?

Line 307 ff.: Also, I think correlation coefficients >0.3 and especially >0.5 cannot be described as “low” (see e.g. (Cohen 1988) where correlations are categorized as small, medium and large for r=0.1, r=0.3 and r=0.5, respectively). Considering the potential high variation in environmental conditions among your forest sites, I think a high significance suggests a relevant effect or co-variation with the respective drivers that should not be down-played. Therefore, I think you have to restate the interpretation of this part of the analysis (you can still emphasize that balsam fir showed higher correlation coefficients than black spruce). (Maybe I misunderstood you? For example you write “In general, correlation coefficients tended to be relatively low even when significant; this was especially true for black spruce as no correlation coefficient between sapling density and gap fraction exceeded 0.5” However, when I look at Table 3, the correlation of sapling density with gap fraction is 0.51***?)

-General comment: The analyses are somewhat descriptive and I was wondering, if these could be complemented by a more systematic analysis of drivers of differences among regeneration clusters?

-So for example, to test the relative importance of different environmental drivers, you could apply a variance partitioning scheme (e.g. “lmg” method (Lindeman et al. 1980) using the R-package by (Grömping 2006)) where the total amount of explained variance can be compared directly among drivers?

-You had this stratified random sampling approach according to “environmental types” but as far as I understand, you did not include this study design variable in your analyses? Did including this variable not affect your results; i.e. do groupings cluster along “environmental types” or not at all?

-k-means clusters’ differences in environmental conditions: It would be interesting to see how robust your findings were:

--For example: Would significant environmental driver differences among clusters change much if you used one or two groups more or less?

--This depends on how the environmental variables’ values were distributed among the k groups. Was there a clear (linear or non-linear) gradient in slope or disturbance in the 2-dimensional space of seedling and sapling densities? Or was every BS or BF cluster a unique combination of environmental conditions? Could you show environmental gradients by colouring the points in the Figures 3 and 4?

5) Discussion: In contrast to the rest of the manuscript, I find the discussion somewhat unclear and speculative in many places; here I would appreciate if authors could stay more closely to their actual results and clearly indicate which statements are potential implications of the results.

Specifically:

Line 363 ff.: Did you test non-linearity or self-organized-ness of your regeneration dynamics? If yes, you would need to explain how and what the results were. Also, what do you mean by self-organized? I would remove these descriptions in the parentheses.

Line 366: It would be more clear if you replaced topographic with “slope”. I would also write “Our analyses suggest that secondary disturbance and topographic constraints are the main drivers”.

Lines 367 ff.: I do not understand why you make the reference to the temporal and spatial scales in this paragraph? As far as I understand you did not test any effects of temporal or spatial scales?

Line 375: As already mentioned above, these Spearman correlations are not necessarily to be classified as low.

Lines 379 ff.: I do not understand why the significant effect of time since last fire is not relevant here?

Lines 406-416 Contain a lot of speculative statements: “Cluster BS7 became cluster BS6”; “and these tree layers are no longer subject to apical control upon the death of the mother tree”; “seedlings benefitted from these openings to produce to a high sapling density, i.e., cluster BS6 shifted to cluster BS5”; “clusters BS2 and BS3 shifted toward cluster BS7, reinitiating the cycle”.

Please make it clear that the progression in regeneration dynamics you describe here are a potential pathway or in accordance or supported or suggested by your results but are NOT actually what you observed in your investigation (you did not follow a forest stand through time or measure apical dominance or investigated how a cycle can be reinitiated, as far as I understand).

Please also elaborate on how certain the suggested pathway is; i.e. are there also other potential pathways consistent with your results?

Line 435: This is a much better framing! “Competition with balsam fir could explain…”

Line 440: Why do you infer that seedling growth is rapid; what is rapid in this case?

Line 443: Why do you infer that there is a return to phase 1 from this point?

Lines 488-490: I do not understand the message of this paragraph.

Line 500: “..stand structure shifts to BF2”; I would restate that this is a suggested pathway.

Lines 514-517: Can you explain this in more detail? I do not see why exactly the successional stages paradigm is clearly refuted? If you write “refuting the classical theory” in the conclusion; this theory has to be properly introduced in the Introduction and assessed in the Methods and Results sections (It would maybe anyway be good to do this, and clearly state where exactly your regeneration dynamics deviate from the classic ones). Also it would be nice if you could add a reference of the classical paradigm.

Line 521 ff.: What do you mean by “trends” in regeneration dynamics? As far as I understand, you did not assess trends, at least not in the sense of changes of regeneration dynamics over time?

Line 527: What do you mean by “treatments must be adapted to conditions within the eastern Canadian forest”?

6) Figures/Tables:

Table 1: Could you add a description when measurements were made? How many times per plot?

Table 2, 4 & 5: Could you indicate the number of study units and indicate if parametric or non-parametric tests were applied?

Fig 1: It would be nice to show the coordinate system on the map. Also it would be nice to see the study design (i.e. the stratified random sampling); e.g. by colouring the dots of the study plots according to environmental types after which they were selected? Were the selected sites a subset from a larger set of sites? Can they also be shown on the map?

Fig 3 & 4: It would be nice to see the distribution of important environmental driver’s gradients as a colour code of the BS & BF cluster points?

Fig 5: Nice paintings!! I find the arrows going up and down indicating the slope a bit confusing. Does this mean there are there no clear gradients of slopes in a specific direction in this two dimensional space of seedling and sapling densities (at least in the case of black spruce)? It would be nice to see the distribution not only of slope but also of secondary disturbance (-symptoms such as gap fraction) across the two dimensional space of seedling and sapling densities. You could also do this by e.g. by colour coding the background or otherwise only one (if necessary, bent) arrow per environmental driver.

MINOR COMMENTS

Abstract:

Line 46: You could clarify the notion of “secondary disturbance” by adding what kind of disturbance you mean (i.e. pathogen outbreaks) and by adding a short description how you assessed it (i.e. by assessing the gap fraction?).

Line 47: You state you assessed effects of “topography” but what you mainly measured was “slope”. 1) To improve the clarity, I would add the notion of slope here or 2) replace the notion of “topography” with the notion of “slope”. (If you do it here, I would do it throughout the manuscript).

Line 49-51: Did you really observe this temporal change in forest dynamics? Or did you rather infer this from the characteristics of the different clusters you identified? Please clearly indicate that you did not measure such a progression but instead that the patterns you find actually suggest such a progression as described.

Line 52-54: Could you please rephrase this statement? I do not exactly understand what is meant here.

Introduction:

Line 64 & 80: Did you mean “anthropogenic”?

Line 127 ff.: Please clarify (2): Do you mean the differences between black spruce and balsam fir regeneration dynamics; OR did you mean the differences between the respective stages of the regeneration process (i.e. “clusters”) for both species?

Materials & Methods:

Line 197: Did you mean “seedling” here?

Conclusion:

Line 518: importance for what?

References

1.

Cohen, J. (1988). Statistical Power Analysis for the Behavioral Sciences. 2nd edn. Lawrence Erlbaum, Hillsdale, NJ.

2.

Grömping, U. (2006). Relative Importance for Linear Regression in R: The Package relaimpo. Journal of Statistical Software, 17, 1-27.

3.

Lindeman, R.H., Merenda, P.F. & Gold, R.Z. (1980). Introduction to Bivariate and Multivariate Analysis. Scott Foresman Glenview, IL.

6. PLOS authors have the option to publish the peer review history of their article (what does this mean?). If published, this will include your full peer review and any attached files.

Reviewer #1: No

---

## [Author Response · Author response to Decision Letter 0]

30 Jun 2020

Reviewer 1

1. Is the manuscript technically sound, and do the data support the conclusions?

Reviewer #1: Partly

2. Has the statistical analysis been performed appropriately and rigorously? 

Reviewer #1: Yes

3. Have the authors made all data underlying the findings in their manuscript fully available?

Reviewer #1: No

4. Is the manuscript presented in an intelligible fashion and written in standard English?

Reviewer #1: Yes

5. Review Comments to the Author

Reviewer #1: SUMMARY

In this study, Martin et al. investigate the patterns and drivers of forest regeneration dynamics in boreal old-growth forests of Canada. Specifically, they assessed stand structural and environmental characteristics of 71 systematically selected forest stands and relate these to differences in black spruce (Picea mariana (L.) Mill.) and balsam fir (Balsam fir (L.) Mill.) regeneration attributes measured by seedling and sapling densities in the years 2015 & 2016. They find that forest regeneration dynamics differ among black spruce and balsam fir species. Nevertheless, in both cases, secondary disturbance and topography were the main drivers explaining regeneration attributes, whereas successional stage and primary disturbance are likely less important. These findings help to shed light on old-growth forest regeneration dynamics and contribute to improving forest management and restoration practices.

GENERAL COMMENTS

The study addresses an important topic, which is not only of high interest to the scientific community, but also to the forest managers and conservationists, as it could guide future silvicultural and restoration practices. In general, the manuscript is well written and methods well described. I like the comparative approach including k-means grouping and a final schematic description of the inferred, general black spruce and balsam fir regeneration processes. Nevertheless, there are some minor shortcomings that need to be addressed before the article can be published.

We are glad the Reviewer has grasped the contribution of our manuscript to forest sciences and the originality of our novel approach. We really appreciate his point of view about our manuscript and we are grateful for the constructive comments provided. In this new version, we have taken into consideration all the suggestions to improve our article. Consequently, we have modified the manuscript in response to the comments to provide a new version. Our individual responses were listed right after particular comments or suggestions. Sentences in bold refers to specific part of the manuscript or of the responses to reviewer, if specified. We strongly believe our manuscript has significantly benefitted from this round of the review process and hope that it is now acceptable for publication in PLOS ONE. In addition, the changes made in the manuscript were also reviewed by a professional proofreader.

1) Concerning “secondary disturbance”: It is not entirely clear to me which kind of disturbance is meant by this notion; could you please more clearly state which processes you refer to in the Introduction? I.e. do you mean pathogen (budworm) outbreaks, windthrow, root rot, fire, and logging? Or do you consider “fire” and “logging” as a primary disturbances?

We provided several clarifications in the introduction to better explain what disturbances we are talking about (outbreaks and windthrows). We provided exampled of secondary disturbances lines 105-106 and we underscored after the objective that stand-replacing disturbance and anthropogenic disturbances were not taken into account in this study (lines 130-132), as they are not related with old-growth dynamics. 

Could explicitly explain in the Methods section how exactly you inferred the secondary disturbance (gap fraction and amount of deadwood,..?) and why these measures are appropriate proxies for secondary disturbance? Finally, it would be great if you could shortly discuss in the Discussion the advantages & disadvantages of approximating secondary disturbances with gap fraction and amount of deadwood?

We now clearly explain in the methods that structural attributes such as high gap fraction, high volume of woody debris, and low basal area can be considered an indicator of recent disturbances (lines 277-283). At the end of the discussion, we added a new paragraph to better discuss the limits in using a synchronous approach to study regeneration dynamics in old-growth forests (lines 620-649). However, we also underlined in this part the advantages of this method given the numerous constraints inherent to this research subject.

Example that needs clarification:

Line 144 ff.: Hence, is fire no secondary disturbance? Is logging considered a secondary disturbance?

We have now clarified that fire is the main natural stand-replacing disturbance on the study territory (Kneeshaw et al. 2011). Even if low-severity fires are observed in Eastern Canada (Kafka et al. 2001), they are considered as relatively rare in comparison to other disturbances such as spruce budworm outbreaks or windthrows (Kneeshaw et al. 2011; Shorohova et al. 2011) and are therefore not mentioned.

We have also clarified lines 130-132 that logging was not taking into account in this study as it is an anthropogenic disturbance whereas we study the impact of natural disturbance regime. Similarly, we specified lines 163-164 that all the studied stands were primary forests, which mean that they have never been logged.

Line 381 ff.: could you please elaborate which types of secondary disturbances you are referring to here?

This part has been deleted with the changes in discussion structure. However, we have been careful to make better mention of the possible types of disturbance in the new parts (e.g., lines 467-471 or 484-485).

Line 470: Please elaborate on the kind of disturbance of balsam fir, which might be different from black spruce?

We have made several changes here to underline that black spruce and balsam fir are driven by the same disturbances but with different sensitivity and/or susceptibility based on the ecology of these study species (lines 567-574).

2) Concerning the notion of “topography”: To approximate “topography” you mainly measured “slope”. To improve the clarity throughout the manuscript, I would add the notion of slope in parentheses after “topography” or entirely replace the notion of “topography” with the notion of “slope”. Also, please explain why you chose “slope” as an important factor representative of “topography” and why you did not test effects of other topographic parameters such as altitude?

We made several changes in the manuscript to better highlight that we used the slope as an indicator of topography (e.g. lines 51-52 and 423-424).

We chose to not include other topographic variables such as the altitude for several reasons. First, the altitudinal range on the study territory is relatively stable, variations in elevation are unlikely to have a significant influence on stands, as would be observed in mountainous regions. On the contrary, it has previously been shown that the slope has a very strong effect on the composition and dynamics of forest stands in the study area (Barrette et al. 2018; Oboite and Comeau 2019). Due to the influence of glaciers on the topography of the study area, the majority of the slopes are exposed either to the east or to the west. Combined with the moderate variations in altitude and slope in the same area, we therefore chose not to include exposure in the study.

Finally, we also considered including TOPEX in our variables to get a better idea of the sensitivity of the study sites to windthrow (Ruel et al. 2002). However, preliminary analyses showed no significant influence. For the sake of concision, we therefore chose not to include it.

3) Methods: Some important aspects of the methods are not clear:

Lines 154 ff.: When did you take measurements? During summers or the whole years of 2015/2016? Once per plot in these 2 years? Or once per plot per year?

Each plot was surveyed one year, in 2015 or 2016. This is now clarified line 162 and 221. More specifically, we made three passes in each of the stands studied during the same year: a first pass to select sites for final sampling (preliminary survey), a main sampling session where most of the variables were measured, and a last pass where some trees were felled to measure their height and collect basal disks. The last passage was done in order to optimize the presence of our colleague authorized to use a chainsaw in isolated areas. However, we do not think it is necessary to give all these minor details in the article to guaranty the reproducibility of this research.

Line 154: Stratified random sampling approach: Did you select from a larger set of possible forest stands? If yes from where (how many sites in total were available)? Also, 71 is not a multiple of 6; therefore not all environmental types are represented with the same amount of replicates? Could you indicate the number and distribution of selected forest stands that correspond to the environmental types? (e.g. you could add 6 different colours instead of the red colour for the dots in Figure 1)?

At the first step of the sampling, we realized an exploratory cartographic analysis based on the aerial surveys performed in 2007 by Quebec government (4th decadal forest survey). This survey divides the forest in different polygons, defined by a homogeneous age and potential vegetation class. It was also the most recent forest survey available on the study territory in 2015 and 2016. We selected all the polygons corresponding to the researched potential vegetations and situated at 200m buffer zone near to the forest roads. At this stage, several hundred of polygons were selected.

Then, we performed the preliminary field survey to identify which polygons were suitable for the final survey (i.e., stands that were not logged nor burned since the 4th decadal forest survey, stands near roads that are still open and for example not destroyed by local floods following the melting of snow, stands generally accessible). We also sampled cores at the base of five trees (two cores per tree) to estimate stand age. Indeed, the maximum age class of the aerial forest surveys is >100 years, which is not precise enough for our requisites (lines 178-183). 

Due to time constraints 94 sites were validated on field during the preliminary survey but only 71 could be eventually sampled, depending on their availability. For this reason, the most isolated sites were generally not sampled, unless they belonged to a hard-to-find class. Similarly, finding stands defined by black spruce–feather moss potential vegetation, gentle slopes, organic deposits, hydric drainage and under 200 years was difficult. This is probably because these stands are less likely to be burned (Madoui et al. 2010) but also less likely to be logged because of the low wood volume. As you can see on the map (Figure 1), we chose to sample a smaller proportion of these sites compared to other potential vegetation to avoid too great an imbalance between the two target age classes. Finally, we used a more precise method to estimate stand age during the final sampling in comparison to preliminary sampling (basal discs vs. cores, 10 trees vs. 5). For this reason, so some of the sampled stands eventually belonged to a different age group than initially expected.

We therefore add new sentences in the methods to better explain some of these different steps (lines 178-191) but we preferred to keep it the shortest possible.

Line 161: What is the “productive forest”? Did you not select from “unproductive forest”? I do not understand the meaning of this sentence.

In Quebec’s typology, land cover can be classified between productive forest areas, unproductive forest areas (i.e., areas with trees but a potential wood volume at 120 years under 30m3/ha, e.g. peatlands) and non-forested areas (e.g., bogs, rocks, lakes). However, we agree that it is not clear for an international readership and we modified it by “The six dominant MFWP environmental types covered more than 72% of the forested area on the study territory” (lines 169-170)

Line 190: Could you explain in more detail how you measured gap length? Did you use hemispheric photography to do this?

This is now explained lines 206-207. Gaps were measured along linear transects using a method defined by (Pham et al. 2004) and adapted by (Martin et al. 2018). Hemispheric photography was unfortunately not used in this sampling.

-Somewhere in this section it would be nice to have a clearer/more structured description that you approximated “topography” by measuring slope using a clinometer, and that you approximated “secondary disturbance” by assessing “gap fraction” and “woody debris”.

This is now clarified lines 221-222 for the topography. For the secondary disturbances, we thought that this part of the method was most suitable in the part “Data analysis” as it is an inherent part of data interpretation (lines 277-283).

Line 210: Did (40) measure the structural attributes in the same time period as the attributes presented in this study?

Yes, to optimize the fieldwork, everything was sampled in the same session, except tree height because we needed to fell the trees first (see lines 172-205 in responses to reviewers). Nevertheless, tree height was sampled the same year than the other attributes. Overall, regeneration data was not used by (Martin et al. 2018), even though it was sampled with the other structural and environmental attributes. We clarified line 221 that all the sampling in a same plot were performed the same year. Thus, we don’t think that it is necessary to precise that regeneration sampling was made simultaneously to the sampling of (Martin et al. 2018). However, if you disagree, we could add this information in the manuscript.

4) Analyses/Results:

Line 257: Could you elaborate a bit more about the significance of the SSI? I.e. what are high/low values? Is a grouping with a SSI of 2.23 much better than a grouping with SSI of 1.5 or only slightly better?

We clarified in the methods the mean of the SSI (lines 261-262), but we don’t think it is necessary to discuss it in detail in the manuscript. The highest SSI value indicate the optimal partition but this doesn’t mean that any other partition is wrong (see lines 357-372 in responses to reviewers). Depending on the context, it can be however interesting to prefer lower SSI values to facilitate the subsequent analyses or interpretation (e.g., if the SSI index keep increasing with the number of clusters, it could be preferable to have a lower number of clusters). As presented lines 357-372 in the responses to reviewers, changing the number of clusters doesn’t significantly change the result but we think it was better to keep the optimal partition, as long as the number of clusters was reasonable.

Line 303 & 322 ff.: This Tables 3, 4 & 5 somehow show results that are not entirely consistent with your main message that “secondary disturbance” and “topography (i.e. slope?)” were the main drivers of regeneration in black spruce and balsam fir:

-e.g. black spruce seedling and sapling densities significantly correlate to cohort basal area proportion & minimum time since last fire (both measures of “succession” as far as I understand[?]) & also with soil organic layer depth. This is also reflected in cluster differences of Table 4. Unless I misunderstood your main message, I think it should be mentioned in the Discussion that in case of black spruce, successional stage and soil organic layer are also important drivers of regeneration dynamics.

-e.g., balsam fir significantly relates to woody debris and snag basal area and not to gap fraction, whereas black spruce relates mainly to gap fraction. Hence the two species are influenced by different secondary disturbance drivers: Could you elaborate on this more in the Discussion?

You are right, these points have not discussed enough. Although we observed a significant correlation between black spruce regeneration and MTSLF or CBAP, the bootstrapped regression indicated a limited influence of these factors (lines 361-374). Similarly, we observed no differences in CBAP and the significant differences in MTSLF doesn’t necessarily indicate different succession stages. As discussed now lines 464-472, this probably means that the lowest CBAP values mean that the oldest trees died and have been recently replaced by younger trees. Indeed, the longevity of black spruce and balsam fir is relatively low (often <200 years), whereas old trees are generally the most susceptible to secondary disturbances such as windthrows or insect outbreaks (Viereck and Johnson 1990; Sturtevant et al. 1997; Morin et al. 2009). This is for these reasons that the age of the oldest trees is not necessarily a relevant factor once stand age exceed 150 years and that the use of the CBAP should be preferred. Overall, the best solution to provide relevant synchronic analyses of old-growth forest dynamics would require extensive C14 analysis of coils in the soil, but unfortunately this is still far too expensive.

For these reasons, when stands are defined by equivalent CBAP values, we think that the study of structural attributes to reconstruct stand dynamics is the best solution available (as discussed now lines 622-649). However, we agree that the choice of this approach and the way we interpreted our results was not discussed enough. For example, we underline now why deadwood volume may be not the most relevant indicator of stand dynamics in black spruce forests lines 478-485. Indeed, due to the relatively thick moss layer in black spruce dominated stands, deadwood can be rapidly buried in the organic horizon (Moroni et al. 2015), a process less common in stands suitable for balsam fir due to the lower thickness of the organic horizon. As such, we observe in general lower deadwood volume in old black spruce stands in the comparison to mixed black spruce -balsam fir stands (Martin et al. 2018). This lower volume can be also reinforced by the size of the trees, which are generally smaller in black spruce-dominated stands (Bergeron and Harper 2009; Martin et al. 2018). Hence, it is likely that a significant fraction of coarse woody debris are retained in the deadwood surveys because of their low size (diameter < 9cm) at the transect intersection.

Line 307 ff.: Also, I think correlation coefficients >0.3 and especially >0.5 cannot be described as “low” (see e.g. (Cohen 1988) where correlations are categorized as small, medium and large for r=0.1, r=0.3 and r=0.5, respectively). Considering the potential high variation in environmental conditions among your forest sites, I think a high significance suggests a relevant effect or co-variation with the respective drivers that should not be down-played. Therefore, I think you have to restate the interpretation of this part of the analysis (you can still emphasize that balsam fir showed higher correlation coefficients than black spruce). (Maybe I misunderstood you? For example you write “In general, correlation coefficients tended to be relatively low even when significant; this was especially true for black spruce as no correlation coefficient between sapling density and gap fraction exceeded 0.5” However, when I look at Table 3, the correlation of sapling density with gap fraction is 0.51***?)

We apology, there is an error in the text. What we meant was that for black spruce, all correlation coefficients were under 0.5, except for the gap fraction. It is possible that this error has passed through the proofreading and revision phase.

We also understand your point about how we can qualify the correlations. In the manuscript, we preferred to remain prudent and not inflate the value of these correlations. The way in which the coefficients are classified can indeed vary greatly from one author to another. For example, based on (Hinkle et al. 2003), correlation coefficients ranging from 0.5 to 0.7 are classified as “moderate”. We therefore provided several changes in this part to better emphasize these correlations.

-General comment: The analyses are somewhat descriptive and I was wondering, if these could be complemented by a more systematic analysis of drivers of differences among regeneration clusters?

-So for example, to test the relative importance of different environmental drivers, you could apply a variance partitioning scheme (e.g. “lmg” method (Lindeman et al. 1980) using the R-package by (Grömping 2006)) where the total amount of explained variance can be compared directly among drivers?

Thank you for proposing this interesting solution. During the preliminary analyses, we tested different methods such as regression or structural equation model to provide more systematic analyses but the results were not convincing and solid-enough in our opinion (the data were too “noisy” to produce satisfying models). However, we didn’t thought about the possibility of using bootstrapping technics to reduce the influence of this noise. In particular, the method you propose provides a confidence interval for each of the parameters analyzed, which allows us to be more careful with the results of the regressions. We therefore add this analysis into the method as a complement to the correlations (methods: lines 271-276, results: lines 358-371, discussion: lines 434-436, Table 4). As you can see, it underscores the absence of any clear and strong driver for black spruce regeneration. In contrast, this analysis underscores the importance of balsam fir proportion in the basal area for balsam fir regeneration.

-You had this stratified random sampling approach according to “environmental types” but as far as I understand, you did not include this study design variable in your analyses? Did including this variable not affect your results; i.e. do groupings cluster along “environmental types” or not at all?

The environmental types were used as a guideline during the sampling to be certain that the stand sampled were representative on the study territory. Indeed, forest roads are built for the sole purpose of providing access to stands for logging in this region. It is therefore not certain that random sampling of forests near roads will ensure that they are representative of the territory. However, we did not choose to include them in the analyses, because we considered that the environmental attributes sampled (slope and thickness of the organic horizon) are more detailed indicators of the characteristics of the plot than potential vegetation. In addition, we considered that comparing the clusters with the potential vegetation would have be too heavy (e.g., a contingency table with 8 clusters and 6 potential vegetation) without providing more details than the results obtained from field data.

-k-means clusters’ differences in environmental conditions: It would be interesting to see how robust your findings were:

--For example: Would significant environmental driver differences among clusters change much if you used one or two groups more or less?

As you can see in the Tables 1 to 3 and Figures 1 to 3 in responses to reviewers, the general patterns observed with respectively 8 clusters for black spruce and 4 clusters for balsam fir don’t significantly change in comparison to the results presented in the manuscript (however, for balsam fir, we preferred to split the clusters in 2 groups rather than 3 or 5 because of the low SSI values observed with these partitions). We think that one of the great advantages of our approach is its overall plasticity when observing complex processes (i.e., processes that cannot be summarized by a linear or near-linear trends based on a small number of parameters), which is the case in old-growth forests, while providing robust results (see also response to reviewers lines 231-240). Admittedly, its main limitation is then the interpretation of the different clusters, which must be done in carefully and requires a fine knowledge of the ecology of the ecosystems studied.

(Please, see the attached file to access the figures)

Figure 1: Scatterplot of black spruce regeneration with 7 clusters

Table 1: Environmental attributes of the 7 clusters of black spruce regeneration

Figure 2: Scatterplot of black spruce regeneration with 9 clusters

Table 2: Environmental attributes of the 9 clusters of black spruce regeneration

Figure 3: Scatterplot of balsam fir regeneration with 2 clusters

Table 3: Environmental attributes of the 2 clusters of balsam fir regeneration

--This depends on how the environmental variables’ values were distributed among the k groups. Was there a clear (linear or non-linear) gradient in slope or disturbance in the 2-dimensional space of seedling and sapling densities? Or was every BS or BF cluster a unique combination of environmental conditions? Could you show environmental gradients by colouring the points in the Figures 3 and 4?

For reasons of consistency, we think it would be relevant in this case to represent how all the variables of importance are distributed at the level of the sampled points. Presenting only the environmental variables might indeed seem strange to the reader.

We therefore propose adding two figures (Figures 5 and 6), showing the values of the different variables that proved significant to Spearman's correlations for black spruce and balsam fir regeneration densities. This way, it provides a panorama that we believe to be more exhaustive.

5) Discussion: In contrast to the rest of the manuscript, I find the discussion somewhat unclear and speculative in many places; here I would appreciate if authors could stay more closely to their actual results and clearly indicate which statements are potential implications of the results.

Specifically:

Line 363 ff.: Did you test non-linearity or self-organized-ness of your regeneration dynamics? If yes, you would need to explain how and what the results were. Also, what do you mean by self-organized? I would remove these descriptions in the parentheses.

You are right, this part doesn’t add relevant information and can be deleted

Line 366: It would be more clear if you replaced topographic with “slope”. I would also write “Our analyses suggest that secondary disturbance and topographic constraints are the main drivers”.

In this context, we would keep topographic constraints but with a precision: slope is an important factor but also the drainage, as indicated by the depth of the organic horizon

Lines 367 ff.: I do not understand why you make the reference to the temporal and spatial scales in this paragraph? As far as I understand you did not test any effects of temporal or spatial scales?

This is now clarified: Temporal scale is indicated by the influence of disturbance regime whereas the spatial scale is represented by the local changes in topography (i.e., slope and depth of the organic horizon) (lines 423-424).

Line 375: As already mentioned above, these Spearman correlations are not necessarily to be classified as low.

This is now corrected (line 431)

Lines 379 ff.: I do not understand why the significant effect of time since last fire is not relevant here?

This is now discussed in details lines 464-472. See also responses to reviewers lines 257-296.

Lines 406-416 Contain a lot of speculative statements: “Cluster BS7 became cluster BS6”; “and these tree layers are no longer subject to apical control upon the death of the mother tree”; “seedlings benefitted from these openings to produce to a high sapling density, i.e., cluster BS6 shifted to cluster BS5”; “clusters BS2 and BS3 shifted toward cluster BS7, reinitiating the cycle”.

Please make it clear that the progression in regeneration dynamics you describe here are a potential pathway or in accordance or supported or suggested by your results but are NOT actually what you observed in your investigation (you did not follow a forest stand through time or measure apical dominance or investigated how a cycle can be reinitiated, as far as I understand).

Please also elaborate on how certain the suggested pathway is; i.e. are there also other potential pathways consistent with your results?

You are right, this part was too speculative, without clearly distinguishing the results from the hypothetical pathways that we deduce from it. This has therefore been largely corrected in the manuscript to be more factual (see Discussion). We have also indicated the possible alternative paths or other dynamics, but also the reasons why they seem less likely to occur (lines 521-531).

Line 435: This is a much better framing! “Competition with balsam fir could explain…”

This section has been changed to better highlight this hypothesis but we also discuss in more detail why it should be not the only point to take into account to explain the results observed (lines 434-458).

Line 440: Why do you infer that seedling growth is rapid; what is rapid in this case?

Indeed, “rapid” is not necessary here. We therefore removed it (lines 531-539).

Line 443: Why do you infer that there is a return to phase 1 from this point?

At this stage, the seedlings have now totally closed the gap. If the stand is not disturbed again in the next years, it means that gap-fillers will continue to grow in size and diameter, increasing the basal area. We also expect that new layers will progressively appear in the understory, as they can grow at this stage due to the absence of new gaps. For this reason, we think that stand structure return to the first phase (relatively high basal area and moderate gap fraction, high seedling density, low sapling density)

Lines 488-490: I do not understand the message of this paragraph.

You are right, it was not very clear. This is now clarified lines 595-601. As spruce budworm outbreaks is a species-specific disturbance (i.e., the mortality in balsam fir is significantly higher than black spruce ; Morin et al. 2009; De Grandpré et al. 2018), it has been therefore hypothesized that the presence of balsam fir in the old-growth stands were black spruce is still competitive can greatly vary in time. 

Line 500: “..stand structure shifts to BF2”; I would restate that this is a suggested pathway.

This is corrected (line 607)

Lines 514-517: Can you explain this in more detail? I do not see why exactly the successional stages paradigm is clearly refuted? If you write “refuting the classical theory” in the conclusion; this theory has to be properly introduced in the Introduction and assessed in the Methods and Results sections (It would maybe anyway be good to do this, and clearly state where exactly your regeneration dynamics deviate from the classic ones). Also it would be nice if you could add a reference of the classical paradigm.

You are right, this was a clumsy phrasing and it has been modified. As discussed lines 652-658, most of the dynamics of boreal old-growth forests has been described along linear temporal chronosequences. Their relevance after a certain period of time since the fire is therefore questionable due to the increasing loss of accuracy associated with this information (lines 464-472 and see responses to reviewers lines 257-296). Hence, we wanted to underscore that the results of this study provide interesting alternatives to these previous models, as we hypothesize the inner-dynamics of old-growth forests under a secondary disturbance regime and not only the process of forest succession.

Line 521 ff.: What do you mean by “trends” in regeneration dynamics? As far as I understand, you did not assess trends, at least not in the sense of changes of regeneration dynamics over time?

“Trends” was incorrect, you are right. This has been replaced by “potential pathways” (lines 662)

Line 527: What do you mean by “treatments must be adapted to conditions within the eastern Canadian forest”?

These silvicultural treatments were previously developed in Europe more than one century ago, while in Canada remains at the beginning step (20 years ago). Thus, many information is lacking about how the response of Canadian forests stands in terms of growth, mortality, regeneration after partial cutting. This, more research is necessary to adapt these practices to the Canadian conditions (species, topography…). In the current version this question was clarified. We have preferred not to go into too much detail in this section in order to remain concise, but we have nevertheless chose to emphasize that near-natural management strategies must take into account the specific characteristics of boreal forests. For example, the risk of windthrow strongly increase when increase the harvested intensity (Bose et al. 2014; Fenton et al. 2014; Montoro Girona et al. 2019). Similarly, a too high proportion of basal area harvested can be detrimental to old-growth-related species (Fenton et al. 2014).

6) Figures/Tables:

Table 1: Could you add a description when measurements were made? How many times per plot?

Since the measurements on each plot were conducted in the same year, we do not believe it is necessary to add this information to the table. Similarly, we do not believe that adding the sampling date adds any relevant information to the map: the methodology used was the same for the two consecutive years. As described lines 161-170 in the responses to reviewers, the final sampling was mostly carried out according to the various accessibility and logistical constraints. Hence, we think that this could reduce the clarity of the map without adding relevant information.

Table 2, 4 & 5: Could you indicate the number of study units and indicate if parametric or non-parametric tests were applied?

This is a good idea. For each attribute, the number of study units as well as the nature of the test is now clarified in Tables 2, 3, 4 and 5

Fig 1: It would be nice to show the coordinate system on the map. Also it would be nice to see the study design (i.e. the stratified random sampling); e.g. by colouring the dots of the study plots according to environmental types after which they were selected? Were the selected sites a subset from a larger set of sites? Can they also be shown on the map?

Due to copyright constraint, we also changed the general background of the maps. We have used these changes to highlight the different potential vegetation on the study sites and to add the coordinates on the map (Figure 1). However, we no longer have the exact locations of the sites during the preliminary surveys. We have indeed taken new GPS points, with a better accuracy, during the final sampling and representing the center of the plots. It is this data that was finally kept in our database.

Fig 3 & 4: It would be nice to see the distribution of important environmental driver’s gradients as a colour code of the BS & BF cluster points?

As discussed lines 395-406 in the responses to reviewers, we think that it would be clearer to add new figures rather than adding colors in these figures (Figures 5 and 6).

Fig 5: Nice paintings!! I find the arrows going up and down indicating the slope a bit confusing. Does this mean there are there no clear gradients of slopes in a specific direction in this two dimensional space of seedling and sapling densities (at least in the case of black spruce)? It would be nice to see the distribution not only of slope but also of secondary disturbance (-symptoms such as gap fraction) across the two dimensional space of seedling and sapling densities. You could also do this by e.g. by colour coding the background or otherwise only one (if necessary, bent) arrow per environmental driver.

Thank you for the compliment, the painter was very happy to know that you liked the illustration! Indeed, changing the shapes of the arrows based on the slopes was maybe quite confusing. For this reason, we propose minor changes in the figure where the emphasis is placed on the processes rather than the topography. We then add comments near to the arrows to better explain the changes observed in the stand structure or environment (in the case of paludification), as well as the characteristics of the slope when necessary. For example, for black spruce regeneration, the slope is important only for clusters BS1 and BS8 (in this case, the low slope results in a low drainage, hence stand paludification). However, for the other clusters, precising the slope is not necessary as the general processes are the same.

Finally, we don’t think that adding colors in the background is suitable because it could make it more difficult to read the figure. We think that the different changes in the arrows and text will clarify the figure while keeping it understandable.

MINOR COMMENTS

Abstract:

Line 46: You could clarify the notion of “secondary disturbance” by adding what kind of disturbance you mean (i.e. pathogen outbreaks) and by adding a short description how you assessed it (i.e. by assessing the gap fraction?).

This is now clarified line 51.

Line 47: You state you assessed effects of “topography” but what you mainly measured was “slope”. 1) To improve the clarity, I would add the notion of slope here or 2) replace the notion of “topography” with the notion of “slope”. (If you do it here, I would do it throughout the manuscript).

As written lines 140-158 in response to reviewers, we have now taken care to clarify what the topography means in the manuscript. For this part of the abstract, we however think it is important that topography can be either represented by the slope or the drainage in specific case (paludification) (lines 51-52)

Line 49-51: Did you really observe this temporal change in forest dynamics? Or did you rather infer this from the characteristics of the different clusters you identified? Please clearly indicate that you did not measure such a progression but instead that the patterns you find actually suggest such a progression as described.

We now precise that the patterns are explained by indirect indices (lines 49-56). However, due to the limit in word numbers for the abstract, we cannot add many details here.

Line 52-54: Could you please rephrase this statement? I do not exactly understand what is meant here.

Done (see lines 49-50).

Introduction:

Line 64 & 80: Did you mean “anthropogenic”?

Thank you for noticing it. This is now corrected lines 67 and 83.

Line 127 ff.: Please clarify (2): Do you mean the differences between black spruce and balsam fir regeneration dynamics; OR did you mean the differences between the respective stages of the regeneration process (i.e. “clusters”) for both species?

What we mean is the first option you presented. To avoid confusion, we can say that what we observe is the "general" regeneration dynamics of the species (lines 134-135).

Materials & Methods:

Line 197: Did you mean “seedling” here?

Yes, it is now corrected (lines 215)

Conclusion:

Line 518: importance for what?

This word is now deleted.

References

1.

Cohen, J. (1988). Statistical Power Analysis for the Behavioral Sciences. 2nd edn. Lawrence Erlbaum, Hillsdale, NJ.

2.

Grömping, U. (2006). Relative Importance for Linear Regression in R: The Package relaimpo. Journal of Statistical Software, 17, 1-27.

3.

Lindeman, R.H., Merenda, P.F. & Gold, R.Z. (1980). Introduction to Bivariate and Multivariate Analysis. Scott Foresman Glenview, IL.

References

Barrette M, Tremblay S, Auger I. 2018. Commercial thinning that maintained species diversity of a mixed black spruce–jack pine stand enhanced productivity. Scand J For Res. 0(0):1–8. doi:10.1080/02827581.2018.1495254. https://www.tandfonline.com/doi/full/10.1080/02827581.2018.1495254.

Bergeron Y, Harper KA. 2009. Old-growth forests in the Canadian boreal: the exception rather than the rule? In: Wirth C, Gleixner G, Heimann M, editors. Old-growth forests: function, fate and value. Ecological. New York: Springer. p. 285–300.

Bose AK, Harvey BD, Brais S, Beaudet M, Leduc A. 2014. Constraints to partial cutting in the boreal forest of Canada in the context of natural disturbance-based management: A review. Forestry. 87(1):11–28. doi:10.1093/forestry/cpt047.

Fenton NJ, Imbeau L, Work T, Jacobs J, Bescond H, Drapeau P, Bergeron Y. 2014. Lessons learned from 12 years of ecological research on partial cuts in black spruce forests of northwestern Québec. For Chron. 89(03):350–359. doi:10.5558/tfc2013-065.

De Grandpré L, Waldron K, Bouchard M, Gauthier S, Beaudet M, Ruel JC, Hébert C, Kneeshaw DD. 2018. Incorporating insect and wind disturbances in a natural disturbance-based management framework for the boreal forest. Forests. 9(8):1–20. doi:10.3390/f9080471.

Hinkle D, W W, Jurs S. 2003. Applied Statistics for the Behavioral Sciences. 5th ed. Mifflin H, editor. Boston.

Kafka V, Gauthier S, Bergeron Y. 2001. Fire impacts and crowning in the boreal forest: study of a large wildfire in western Quebec. 10.

Kneeshaw DD, Bergeron Y, Kuuluvainen T. 2011. Forest ecosystem structure and disturbance dynamics across the circumboreal forest. In: Millington A, editor. Handbook of biogeography. London. p. 261–278.

Madoui A, Leduc A, Gauthier S, Bergeron Y. 2010. Spatial pattern analyses of post-fire residual stands in the black spruce boreal forest of western Quebec. Int J Wildl Fire. 19(8):1110–1126. doi:10.1071/WF10049.

Martin M, Fenton NJ, Morin H. 2018. Structural diversity and dynamics of boreal old-growth forests case study in Eastern Canada. For Ecol Manage. 422(April):125–136. doi:10.1016/j.foreco.2018.04.007. http://linkinghub.elsevier.com/retrieve/pii/S0378112718301257.

Montoro Girona M, Morin H, Lussier J-M, Ruel J-C. 2019. Post-cutting mortality following experimental silvicultural treatments in unmanaged boreal forest stands. Front For Glob Chang. 2(March):16. doi:10.3389/FFGC.2019.00004. https://www.frontiersin.org/articles/10.3389/ffgc.2019.00004/abstract.

Morin H, Laprise D, Simon AA, Amouch S. 2009. Spruce budworm outbreak regimes in in eastern North America. In: Gauthier S, Vaillancourt M-A, Leduc A, Grandpré L De, Kneeshaw DD, Morin H, Drapeau P, Bergeron Y, editors. Ecosystem management in the boreal forest. Québec: Les Presses de l’Université du Québec. p. 156–182.

Moroni MT, Morris DM, Shaw C, Stokland JN, Harmon ME, Fenton NJ, Merganičová K, Merganič J, Okabe K, Hagemann U. 2015. Buried Wood: A Common Yet Poorly Documented Form of Deadwood. Ecosystems. doi:10.1007/s10021-015-9850-4. http://link.springer.com/10.1007/s10021-015-9850-4.

Oboite FO, Comeau PG. 2019. Competition and climate influence growth of black spruce in western boreal forests. For Ecol Manage. 443(April):84–94. doi:10.1016/j.foreco.2019.04.017. https://doi.org/10.1016/j.foreco.2019.04.017.

Pham AT, De Grandpré L, Gauthier S, Bergeron Y. 2004. Gap dynamics and replacement patterns in gaps of the northeastern boreal forest of Quebec. Can J For Res. 34(2):353–364. doi:10.1139/x03-265. http://www.nrcresearchpress.com/doi/abs/10.1139/x03-265.

Ruel JC, Mitchell SJ, Dornier M. 2002. A GIS based approach to map wind exposure for windthrow hazard rating. North J Appl For. 19(4):183–187. doi:10.1093/njaf/19.4.183.

Shorohova E, Kneeshaw DD, Kuuluvainen T, Gauthier S. 2011. Variability and dynamics of old- growth forests in the circumboreal zone: implications for conservation, restoration and management. Silva Fenn. 45(5):785–806.

Sturtevant BR, Bissonette JA, Long JN, Roberts DW, Applications SE, May N. 1997. Coarse Woody Debris as a Function of Age , Stand Structure , and Disturbance in Boreal Newfoundland. Ecol Appl. 7(2):702–712.

Viereck LA, Johnson WF. 1990. Picea mariana (Mill) B.S.P. — Black Spruce. In: Service USDAF, editor. Silvics of North America, Vol.1 Conifers. Washington D.C. p. 227–237.

---

## [Editor Report · Decision Letter 1]

6 Jul 2020

Driving factors of conifer regeneration dynamics in eastern Canadian boreal old-growth forests

PONE-D-20-04997R1

Dear Dr. Martin,

We’re pleased to inform you that your manuscript has been judged scientifically suitable for publication and will be formally accepted for publication once it meets all outstanding technical requirements.

Kind regards,

RunGuo Zang

Academic Editor

PLOS ONE

Additional Editor Comments (optional):

Accept
---

## [Editor Report · Acceptance letter]

9 Jul 2020

PONE-D-20-04997R1 

Driving factors of conifer regeneration dynamics in eastern Canadian boreal old-growth forests 

Dear Dr. Martin:

I'm pleased to inform you that your manuscript has been deemed suitable for publication in PLOS ONE. Congratulations! Your manuscript is now with our production department. 

Kind regards, 

on behalf of

Professor RunGuo Zang 

Academic Editor

PLOS ONE